# Phototropin connects blue light perception to starch metabolism in green algae

Yizhong Yuan [1,11], Anthony A. Iannetta[2], Minjae Kim [3], Patric W. Sadecki[2], Marius Arend [4,5,6], Angeliki Tsichla[1,12], M. Águila Ruiz-Sola [1,13], Georgios Kepesidis[1,14], Denis Falconet[1], Emmanuel Thevenon[1], Marianne Tardif [7], Sabine Brugière[7], Yohann Couté [7], Jean Philippe Kleman [8], Irina Sizova[9], Marion Schilling [1], Juliette Jouhet[1], Peter Hegemann [9], Yonghua Li-Beisson [3], Zoran Nikoloski [4,5,6], Olivier Bastien [1], Leslie M. Hicks[2] & Dimitris Petroutsos [1,10] ✉

In photosynthetic organisms, light acts as an environmental signal to control their development and physiology, as well as energy source to drive the conversion of $CO_2$ into carbohydrates used for growth or storage. The main storage carbohydrate in green algae is starch, which accumulates during the day and is broken down at night to meet cellular energy demands. The signaling role of light quality in the regulation of starch accumulation remains unexplored. Here, we identify PHOTOTROPIN-MEDIATED SIGNALING KINASE 1 (PMSK1) as a key regulator of starch metabolism in *Chlamydomonas reinhardtii*. In its phosphorylated form (PMSK1-P), it activates GLYCERALDEHYDE-3-PHOSPHATE DEHYDROGENASE (GAP1), promoting starch biosynthesis. We show that blue light, perceived by PHOTOTROPIN, induces PMSK1 dephosphorylation that in turn represses GAP1 mRNA levels and reduces starch accumulation. These findings reveal a previously uncharacterized blue light-mediated signaling pathway that advances our understanding of photoreceptor-controlled carbon metabolism in microalgae.

Photosynthetic microalgae convert light into chemical energy in the form of ATP and NADPH, which fuel $CO_2$ fixation in the Calvin–Benson cycle. This process of $CO_2$ fixation is initiated by the activity of the $CO_2$-assimilating enzyme ribulose 1,5-bisphosphate carboxylase/oxygenase (Rubisco)[1]. In eukaryotic algae, such as the model photosynthetic green microalga *Chlamydomonas reinhardtii* (hereafter *Chlamydomonas*), concentrated $CO_2$ is delivered to Rubisco within a specialized microcompartment in the chloroplast called the pyrenoid[2].

Light is also a spatiotemporal signal; red light is detected by bilin-containing phytochromes, while blue light is perceived by flavin-

[1]Université Grenoble Alpes, CNRS, CEA, INRAE, IRIG-LPCV, Grenoble, France. [2]Department of Chemistry, University of North Carolina at Chapel Hill, Chapel Hill, NC, USA. [3]Institute de Biosciences et Biotechnologies Aix-Marseille, Aix Marseille University, CEA, CNRS, BIAM, Saint Paul-Lez-Durance, France. [4]Bioinformatics Department, Institute of Biochemistry and Biology, University of Potsdam, Potsdam, Germany. [5]Systems Biology and Mathematical Modeling Group, Max Planck Institute of Molecular Plant Physiology, Potsdam, Germany. [6]Bioinformatics and Mathematical Modeling Department, Center of Plant Systems Biology and Biotechnology, Plovdiv, Bulgaria. [7]Université Grenoble Alpes, INSERM, CEA, UA13 BGE, CNRS, CEA, FR2048, Grenoble, France. [8]Université Grenoble Alpes, CNRS, CEA, IBS, Grenoble, France. [9]Institute of Biology, Experimental Biophysics, Humboldt-Universität zu Berlin, Berlin, Germany. [10]Department of Organismal Biology, Uppsala University, Uppsala, Sweden. [11]Present address: Department of Plant Biology, Carnegie Institution for Science, Stanford, CA, USA. [12]Present address: Department of Organismal Biology, Uppsala University, Uppsala, Sweden. [13]Present address: Instituto de Bioquímica Vegetal y Fotosíntesis, Consejo Superior de Investigaciones Científicas-Universidad de Sevilla, Sevilla, Spain. [14]Present address: Sandia National Laboratories, Livermore, CA, USA. ✉e-mail: dimitris.petroutsos@ebc.uu.se

containing cryptochromes and/or PHOTs, the latter characterized by two photosensory light, oxygen, or voltage (LOV) domains[3]. *Chlamydomonas* notably lacks phytochromes in its genome[4], nevertheless, this microalga has evolved a rich repertoire of photoreceptors, including a single-copy PHOT, four cryptochromes, eight rhodopsin-like proteins, and the UV-B photoreceptor UVR8[5]. Through this network of specialized photoreceptors, *Chlamydomonas* regulates important cellular functions, including: gene expression, sexual life cycle, phototaxis, and photoprotection[6–13].

Fixed $CO_2$ in combination with nitrogen is used to synthesize amino acids−the building blocks of proteins that drive biochemical reactions. It is also employed in the synthesis of cellular reserves that ensures carbon and energy supply during the waning period. The most abundant carbon reserve in the model photosynthetic green microalga *Chlamydomonas* is starch; its synthesis occurs during the day and its degradation is triggered at night to sustain energy-demanding cellular functions[14]. Starch also serves another critical role as starch sheaths encasing the pyrenoid act as a barrier, reducing $CO_2$ leakage from this structure[15]. While actively dividing, *Chlamydomonas* cells accumulate starch mostly around the pyrenoid. In contrast, when subjected to a stress such as nutrient limitation, starch granules are massively accumulating in the chloroplast stroma[16]. Little is known about the molecular mechanisms underlying the control exerted by light on carbon storage in microalgae, and current knowledge is limited to factors impacting starch accumulation under adverse environmental conditions, such as nitrogen[17] and phosphorus[18] limitation.

A link between light perception and starch accumulation has been suggested in vascular plants. For instance, *Arabidopsis* mutants devoid of the red/far-red photoreceptor phytochrome B have impaired carbon partitioning and although they have reduced $CO_2$ uptake, they over-accumulate daytime sucrose and starch at the expense of growth[19]. Further, the blue light receptor PHOTOTROPIN (PHOT) has been found to mediate starch degradation in guard cells in the light, thus energizing stomatal opening in *Arabidopsis*[20]. Despite early findings reporting that light quality impacts carbohydrate accumulation and metabolism in green algae[21,22], the underlying molecular mechanism connecting light perception to starch accumulation remains unexplored. Here we address this gap by combining genetics, proteomics, and phosphomimetics to unveil a signaling cascade linking blue light perception by PHOT with starch accumulation in *Chlamydomonas*. Phot-dependent de-phosphorylation at a single serine residue of phototropin-mediated signalling kinase 1 (PMSK1) represses GAP1 (also known as GAPDH; glyceraldehyde-3-phosphate dehydrogenase), which we found to act as an enhancer of starch metabolism.

## Results

### The *phot* mutant is a starch hyperaccumulator

We compared starch levels in wild-type CC125 cells (WT) exposed to white and red light and found that red light favored starch accumulation, in agreement with prior work[21]. Interestingly, when low fluence blue light was superimposed on red light, the beneficial effect of red light was lost and cells accumulated starch levels similar to those of cells exposed to white light (Supplementary Fig. 1). Therefore, we reasoned that blue light acts as a repressor of starch accumulation likely via a blue light receptor. To test this hypothesis, we measured starch content in WT and mutant cells lacking different blue-light receptors, including: *acry*, lacking the ANIMAL-TYPE CRYPTOCHROME[6], generated in this study, *pcry* lacking the PLANT-TYPE CRYPTOCHROME[23], generated in this study, the double *acrypcry* mutant, generated in this study, and the *phot* mutant, devoid of PHOTOTROPIN (generated in ref. 24). When grown asynchronously under continuous light, the starch of the cryptochrome mutants was indistinguishable from that of the WT (Fig. 1a) and the same was true when cultures were synchronized under 12 h light/12 h dark cycles (Supplementary Fig. 2), favoring

cryptochrome accumulation in *Chlamydomonas*[6,23]. In contrast to the cryptochrome mutants, the *phot* mutant accumulated approximately three times more starch than the WT under continuous illumination (Fig. 1a). This phenotype was rescued by ectopic expression of the WT *PHOT* gene; it was fully rescued in strain *phot-C1*[25] (Fig. 1b), which accumulates PHOT protein at WT levels (Supplementary Fig. 3a), and partially rescued in strain *phot-C2* (Fig. 1b), which accumulates PHOT protein but to a lesser extent than the WT (Supplementary Fig. 3a). Complete rescue of the phenotype was also achieved by complementation of a permanently active *PHOT* mutant lacking the LOV sensory domains (*phot-kin* strain; Fig. 1b and Supplementary Fig. 3a, b); a *phot-kin* strain in a different genetic background has previously been shown to exhibit PHOT kinase activity in a light-quality-independent manner[13]. The accumulation of high levels of starch in the *phot* mutant was also confirmed by transmission electron microscopy (TEM), revealing striking changes in starch deposition patterns compared to the wild type. As indicated by as the arrows in the TEM images in Fig. 1c, the chloroplasts of the *phot* mutant are filled with starch granules and its pyrenoid, is surrounded by abnormally thick starch sheaths. In contrast, in the chloroplasts of the WT and *phot-C1* strains starch is predominantly localized as thin sheaths around the pyrenoid (Fig. 1c and Supplementary Fig. 4).

We further explored the link between PHOT and starch accumulation in synchronized cultures under white, blue or red light. Starch accumulated during the light phase and degraded during the dark phase, in accord with[14,26], under all light qualities and in all different strains tested (Fig. 1d). We found that starch levels in WT are higher under red light, where PHOT is inactive, than under white or blue, where PHOT is active (Fig. 1d). In the *phot* mutant, starch levels are high in all three light qualities, while the fully complemented *phot-C1* behaves like WT and the partially complemented *phot-C2* is intermediate between WT and *phot*. In contrast to the *phot* mutant, starch levels in strain *phot-kin*, in which PHOT is active, are low and unaffected by the light quality (Fig. 1d). These data (Fig. 1) indicate that PHOT acts as suppressor of starch accumulation in *Chlamydomonas*. We provided further evidence for this claim by using another set of *phot* mutants in a different genetic background, the *cw15-302*[8,13]. In accordance with the data in Fig. 1, we found that the *cw15-302 phot* mutant accumulated more starch than *cw15-302* (Supplementary Fig. 3c). This phenotype was only partially rescued in the complemented *pphot* strain, which accumulates lower levels of PHOT protein than the *cw15-302* WT strain[13], and was fully rescued in the *cw15-302 phot* mutant expressing the PHOT kinase domain (*pkin* strain; Supplementary Fig. 3c). Conversely, complementation of *cw15-302 phot* cells with truncated gene carrying only the photosensory domains LOV1 and 2 (*plov* strain) did not rescue starch accumulation. Importantly, we observed a similar phenotype when the *cw15-302 phot* mutant was complemented with the dead kinase PHOT (*pkin-D* strain; Supplementary Fig. 3c), demonstrating that the suppression of starch accumulation by PHOT requires its kinase activity. Growth (Supplementary Fig. 5a) and photosynthesis (Supplementary Fig. 5b) were not affected by the overaccumulation of starch in *phot* (Fig. 1c). Finally, *phot* showed no difference to WT in terms of total protein (Supplementary Fig. 6), lipid content (Supplementary Fig. 7) and composition (Supplementary Fig. 8).

### GAP1 is a key regulator of starch metabolism and is controlled by PHOT

To gain more insight into the molecular mechanism underlying the PHOT-mediated light perception and starch accumulation, we applied mass spectrometry (MS)-based quantitative proteomics to compare WT and *phot* cells grown in asynchronous photoautotrophic conditions under continuous white light. Gene ontology enrichment analyses revealed that carbohydrate metabolic processes are upregulated in the *phot* mutant (Fig. 2a). We notably found

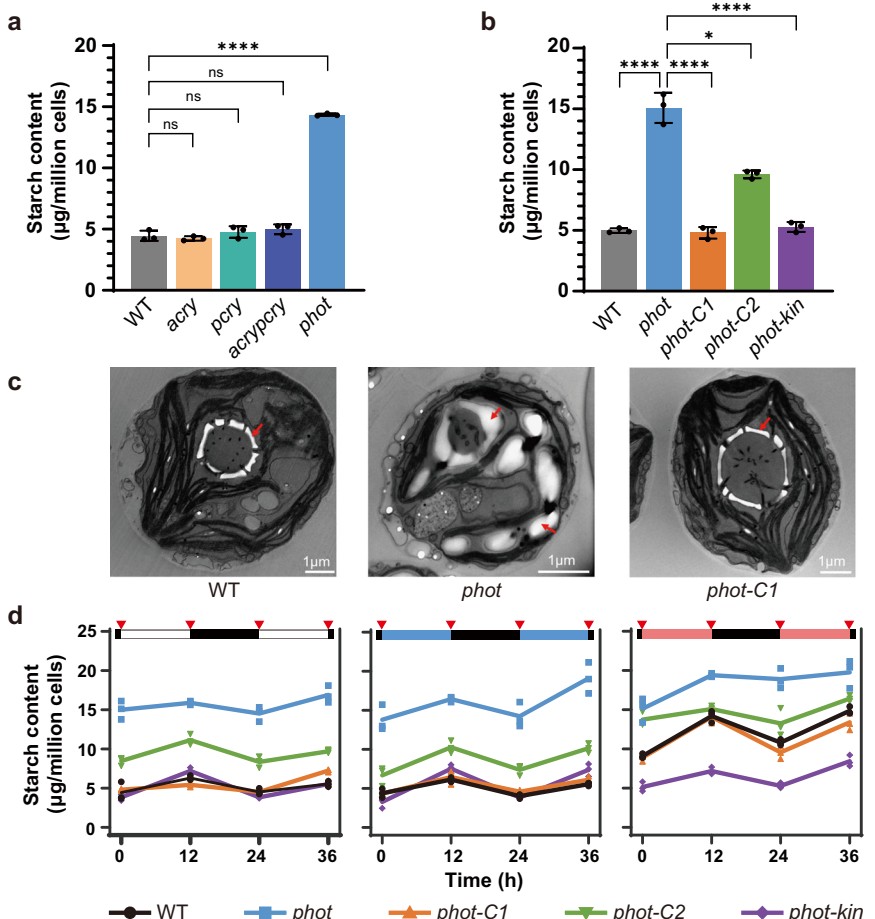

**Fig. 1 | PHOT inhibits starch accumulation in *Chlamydomonas reinhardtii*.**
**a** Starch content of WT, *acry, pcry, acrypcry* lines under continuous light. **b** Starch content of WT and variously *phot*-complemented lines under continuous white light. **c** Transmission electron microscopy pictures of WT, *phot* and *phot-C* synchronized to a12/12 dark/light cycle. Samples were collected at the end of the light phase. Red arrows indicate starch granules. Representative images from three replicates were shown. **d** Starch content of WT and *phot*-complemented lines synchronized to 12/12 dark/light cycle. Red triangles indicate sample collection time. The dark phase is indicated by black bars above the graphs; the light phase by white, blue or red bars, depending on the light quality used. In some cases, the error bars are smaller than the data point symbols. Data are represented as mean ± SD (n = 3 biologically independent samples). The statistical significance was determined using one-way ANOVA with Dunnett's multiple comparisons tests (**a**, **b**) and two-way ANOVA with Dunnett's multiple comparisons tests (**d**). Asterisks indicated the p-values (*p < 0.05; ****p < 0.0001; ns, not significant). In some cases, the error bars are smaller than the data point symbols. Detailed statistical analyses are presented in the Source Data File.

that GLYCERALDEHYDE-3-PHOSPHATE DEHYDROGENASE (GAP1) was expressed 27.5-fold higher in *phot* compared to WT (Fig. 2a, Supplementary Data 2). GAP1 is a chloroplast isoform of GAPDH in *Chlamydomonas*, a Calvin-Benson cycle enzyme predominantly localized in the stromal region surrounding the pyrenoid[27,28]. The significant accumulation of GAP1 in *phot* was in accord with the high accumulation of *GAP1* mRNA (Fig. 2b). While mRNA accumulation of all tested starch-related genes was high in the *phot* mutant in the middle of the light phase (Supplementary Fig. 9), *GAP1* was the most highly expressed, prompting further investigation into the link between PHOT and GAP1. We found that *GAP1* mRNA overaccumulation phenotype of *phot* was completely rescued in *phot-C1* and *phot-kin* and partially rescued in *phot-C2* (Fig. 2b), in accord with the PHOT expression levels in the different complemented lines (Supplementary Fig. 3a, b), suggesting that PHOT acts as suppressor of *GAP1* upon illumination. Similar to the PHOT-mediated suppression of starch (Supplementary Fig. 3c), the PHOT-dependent suppression of *GAP1* requires the kinase activity of PHOT (Supplementary Fig. 10). The *pkin-D* strain, a *phot* mutant expressing a dead kinase PHOT with point mutations in the ATP binding site[8] and the *plov* strain expressing only the sensory domains LOV, both

accumulate *GAP1* mRNA at the level of *phot*. Complementation of the *phot* mutant with the full-length PHOT (*phot* strain) or its kinase domain (*pkin* strain) rescues *GAP1* mRNA to WT levels (Supplementary Fig. 10).

We also measured *GAP1* mRNA in cells synchronized to a 12/12 light-dark cycle. *GAP1* mRNA in WT was found to be strongly influenced by the diurnal cycle, starting at very low levels after the initial onset of white light, rising to maximal levels at the end of the light phase, and then progressively and slightly declining during the night phase (Fig. 2c); this is in accordance with data from previous studies[29,30] on the impact of diel cycle on the genome-wide gene expression (replotted in Supplementary Fig. 11). *GAP1* mRNA accumulation profile was identical in white and blue light conditions. However, under red light, expression levels at the beginning of the day were significantly bigger and reached approximately 45 times higher levels at the midpoint of the light phase (Fig. 2c), suggesting that the low levels of *GAP1* mRNA after the initial onset of light are a result of suppression by blue light. Indeed, in all light qualities tested, *GAP1* mRNA levels in the synchronized *phot* mutant (Fig. 2d) were identical to those of the WT in red light (Fig. 2c). The fully complemented *phot-C1* behaved like WT (Fig. 2e) while the partially complemented *phot-C2* only partially

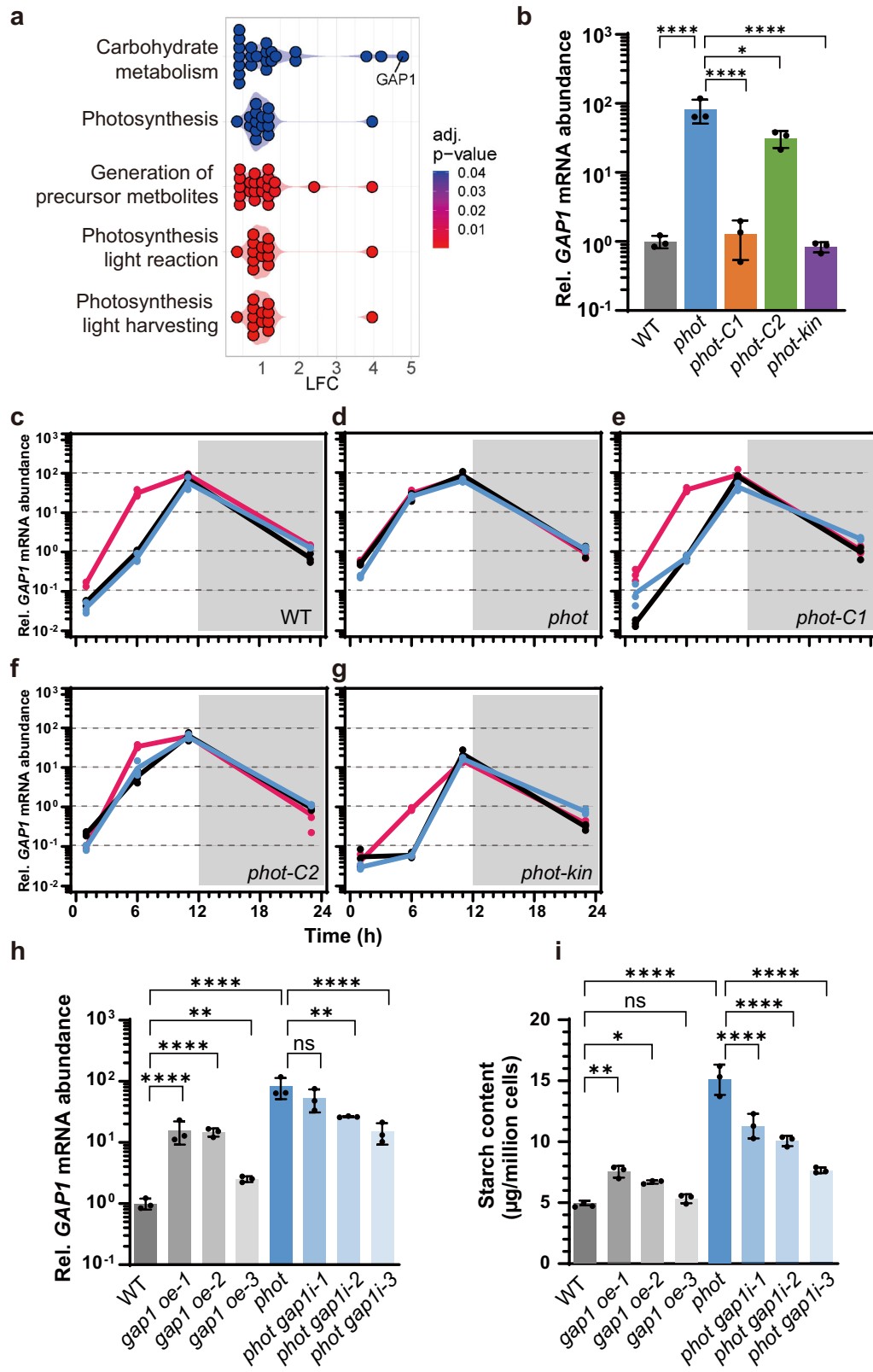

rescued the observed phenotype (Fig. 2f). Notably, the *phot-kin* strain in which PHOT is devoid of the photosensory LOV domains and is always active regardless of light quality[13], accumulated very low levels of *GAP1* mRNA (Fig. 2g; note the log scale). However, in *phot-kin* cells grown under red light, GAP1 mRNA levels are higher at 6 h compared to cells grown in white or blue light (Fig. 2g). Despite, the increased *GAP1* mRNA in red light this was not sufficient for overaccumulation of

starch (Fig. 1d). This indicates that additional regulators, beyond PHOT, may be involved in repressing *GAP1* in a blue light-dependent manner.

Our results so far show a strong association between starch and *GAP1* mRNA accumulation in the different *phot* mutants (Figs. 1b and 2b). WT, *phot-C1* and *phot-kin* accumulate low amounts of starch (Fig. 1b) and low mRNA *GAP1* (Fig. 2b), *phot* accumulates high

**Fig. 2 | GAP1 is inhibited by blue light via PHOT and plays key role in PHOT-dependent starch metabolism in *Chlamydomonas*. a** Gene ontology enrichment analyses for differentially abundant proteins in *phot* compared to WT based on whole cell proteomics data. LFC, log2 fold change (enriched GO terms were only found for proteins with increased abundance in *phot* in comparison to WT); color code indicates adjusted p-value of GO set enrichment (null hypothesis: number of differentially abundant proteins in GO set is hypergeometric random distributed. P-value were adjusted according to Benjamini-Hochberg procedure **b** *GAP1* relative mRNA abundance of WT and *phot*-complemented lines under continuous white light. **c–g** relative mRNA abundance of *GAP1* in WT and *phot*-complemented lines synchronized to a 12/12 dark/light cycle. Phases are indicated by white and gray shading. Line colors indicated light qualities. Black, white light; Red, red light; Blue, blue light. The *GAP1* transcription level (**h**) and starch content (**i**) of WT, *GAP1* overexpression, and *photgap1* knockdown lines under continuous light. Data are presented as mean ± SD (n = 3 biologically independent samples). The statistical significance was determined using one-way ANOVA with Dunnett's multiple comparisons tests (**b**, **h**) and two-way ANOVA with Dunnett's multiple comparisons tests (**c–g**) of log10 transformed mRNA data as indicated in the graphs. The statistical significance was determined using one-way ANOVA with Dunnett's multiple comparisons tests (**i**). Asterisks indicated the p-values (*p < 0.05; **p < 0.01; ****p < 0.0001; ns, not significant). In some cases, the error bars are smaller than the data point symbols. Detailed statistical analyses are presented in the Source Data File.

levels of starch (Fig. 1b) and high *GAP1* mRNA levels (Fig. 2b), and finally the partially complemented *phot-C2* (Supplementary Fig. 3a) accumulates intermediate levels of starch and *GAP1* mRNA (Fig. 1b and Fig. 2b). To further confirm the positive connection between *GAP1* expression level and starch amount, we generated WT strains overexpressing *GAP1* (Supplementary Fig. 12). Strains *gap1-oe1* and *gap1-oe2* accumulated approximately 15-fold more *GAP1* mRNA than WT (Fig. 2h), resulting in 1.5-fold higher starch content (Fig. 2i). The strain *gap1-oe3*, with a more modest overexpression of *GAP1* mRNA (2.5-fold compared to WT; Fig. 2h), had starch content at WT levels (Fig. 2i). For comparison, we included the *phot* mutant in these analyses, which expressed 80-fold more *GAP1* and accumulated 3-fold more starch than the WT. We also downregulated *GAP1* in the *phot* mutant and generated strains *phot-gap1-i1*, *2* and *3* that accumulated 1.6-, 3.1- and 5.5-fold less *GAP1* mRNA (Fig. 2h) and 1.3-, 1.5- and 2-fold less starch, respectively, compared to *phot* (Fig. 2i). In conclusion, overexpression of *GAP1* in WT led to overaccumulation of starch whereas downregulation of *GAP1* in *phot* decreased starch accumulation (Fig. 2h, i). These results (Fig. 2) strongly suggest that GAP1 plays a key role in starch metabolism in *Chlamydomonas*, acting under the control of PHOT.

## PHOT alters phosphorylation state of PHOTOTROPIN-MEDIATED SIGNALLING KINASE 1

To identify missing components in the phototropin-mediated signaling pathway that suppress starch accumulation in *Chlamydomonas*, we compared the phosphoproteome of WT and *phot* cells after an overnight dark acclimation and a 5-min exposure to blue light. We identified 1119 phosphopeptides, belonging to 747 phosphoproteins, and applied unsupervised hierarchical clustering of z-transformed relative abundance of phosphopeptides to obtain an overview of the condition-specific and phototropin-dependent phosphorylation changes (Supplementary Fig. 13 and Supplementary Data 3). Phosphopeptides falling within cluster "E" were highly phosphorylated in the dark and became de-phosphorylated after blue light illumination, but only in the case of WT; in the *phot* mutant they remained highly phosphorylated after blue light illumination. One such peptide was the ADGVSpSPHELTR, phosphorylated at serine120 (S120), which belongs to the gene product of Cre16.g659400 encoding a Ser/Thr protein kinase (Fig. 3b), localized in the cytosol, plasma membrane and flagella (Supplementary Fig. 14). We named this kinase phototropin-mediated signalling kinase 1 (PMSK1).

To validate the phosphoproteomic findings, we generated WT and *phot* lines expressing PMSK1 fused to a FLAG epitope. These lines (WT/*PMSK1-FLAG* and *phot*/*PMSK1-FLAG*) were dark acclimated prior exposure to red or blue light and samples were taken at 5 min intervals for a period of 20 min to assess the PMSK1-FLAG phosphorylation status using Phos-tag SDS-PAGE, followed by immunodetection against FLAG (Supplementary Fig. 15). In accordance with the role of PHOT in the light-dependent dephosphorylation of PMSK1 detected in our phosphoproteomic analyses (Fig. 3a), the Phos-tag data revealed a rapid dephosphorylation of PMSK1-FLAG (Fig. 3c) upon exposure to

light. In the WT background, dephosphorylation occurred only under blue light (Fig. 3c). In contrast, in the *phot* background, dephosphorylation was barely observed, even under blue light (Fig. 3c). Furthermore, no dephosphorylation was observed under red light in either strain (Fig. 3c). These data (Fig. 3a–c) support our hypothesis that dephosphorylation of PMSK1 requires blue light-activated PHOT. Interestingly, the observed blue light dependent mobility shift of PMSK1-FLAG (Fig. 3c) was abolished when S120 was substituted by an alanine (A) or an aspartic acid (D) residue (Fig. 3d). These data indicate that phosphorylation at residue S120 is essential for the shift. However, this shift may not be solely attributed to S120 phosphorylation, but rather to additional phosphorylation events that depend on the initial phosphorylation of S120.

## Phosphorylation state of serine 120 of PMSK1 controls starch accumulation

We next set out to investigate the functional significance of the phosphorylation status of S120 in vivo, taking advantage of the phosphomimetic mutation S120D and the non-phosphorylatable mutation S120A in PMSK1-FLAG, expressed in WT and in *phot* (Supplementary Fig. 15). We measured *GAP1* mRNA and starch accumulation in WT, *phot* and all above-mentioned generated mutants, synchronized under white, blue or red light in a 12 h light/12 h dark regime. Overexpression of PMSK1^S120D^-FLAG in WT, resulted in an enhanced *GAP1* expression level and increased starch content under all three light qualities, exceeding those recorded in the *phot* mutant (Fig. 4a, b and Supplementary Figs. 16 and 17). Overexpression in *phot* of PMSK1^S120D^-FLAG resulted in an even further higher *GAP1* expression level and increased starch content as compared to *phot* (Fig. 4a, b and Supplementary Fig. 17). Conversely, overexpression in *phot* or WT of PMSK1^S120A^-FLAG, resulted in low, WT-level or even lower, *GAP1* mRNA and starch levels, across all three light quality tests (Fig. 4a, b and Supplementary Figs. 16 and 17). We also analyzed the impact of overexpression of the unmodified PMSK1-FLAG in both WT and *phot* mutant (Fig. 4a, b and Supplementary Figs. 15 and 17); WT and WT/*PMSK1-FLAG* behaved very similarly in all light colors with respect to *GAP1* mRNA and starch accumulation (Fig. 4a, b and Supplementary Fig. 17). However, *phot*/*PMSK1-FLAG* showed increased *GAP1* mRNA and accumulated more starch (Fig. 4a, b and Supplementary Fig. 17). This increase is likely due to the absence of PHOT, causing both endogenous and transgenic PMSK1-FLAG in the cells to remain phosphorylated (Fig. 3c).

Taken together, our data (Fig. 4) reveal that the phosphorylation state of S120 of PMSK1 controls starch metabolism through regulation of GAP1 mRNA levels. To investigate if PMSK1's role is dependent on its kinase activity, we used a kinase-inactive version of PMSK1, where Asp-442 was substituted with Asn (D442N) to disrupt the ATP-binding site, alongside the phosphomimetic S120D or the non-phosphorylatable S120A mutations. Neither mutation (S120D or S120A) affected starch content (Supplementary Fig. 18), indicating that PMSK1 fulfills the observed regulatory role on starch metabolism through its kinase activity.

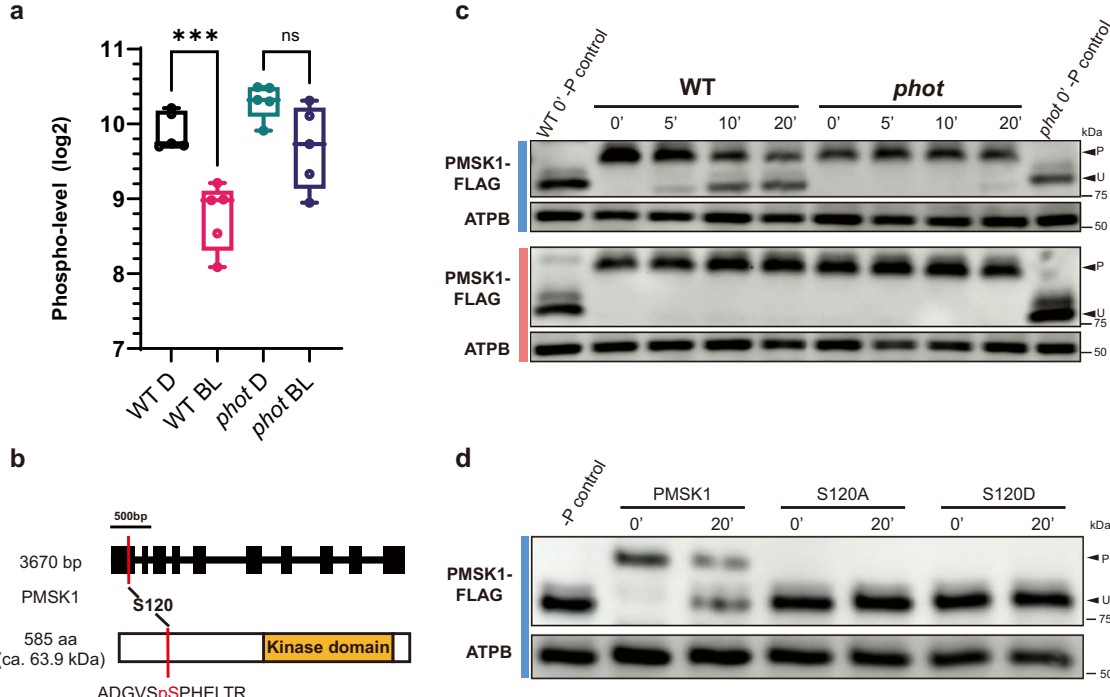

**Fig. 3 | Identification of PMSK1 as a key protein on PHOT-dependent starch metabolism in *Chlamydomonas reinhardtii*. a** Changes in the phosphorylation level of S120 of PMSK1 in response to BL in WT and *phot*, quantified by phospho-proteomics. Samples were collected after 24 h of acclimation to darkness and 5 min after the start of blue light. D dark, BL blue light. Data are presented as log2-transformed phospho-levels (n = 5 biologically independent samples). Error bars indicate mean ± SEM. Box plots represent the median (center line), interquartile range (bounds of the box), minima, and maxima (whiskers) with individual data points overlaid. The statistical significance was determined using one-way ANOVA with Dunnett's multiple comparisons tests. Asterisks indicated the p-values (****p < 0.0001; ns not significant). **b** Upper: Genomic structures of PMSK1. Black boxes and lines indicate exons and introns, respectively. Lower: Schematic structures of PMSK1. Red lines indicate the phosphorylated residue in PMSK1. The orange box indicates the kinase domain. **c** Phosphorylation level of PMSK1-FLAG as a function of time in WT/*PMSK1-FLAG* and *phot*/*PMSK1-FLAG*, exposed to blue or red light after 24 h acclimation to darkness (indicated as t = 0 in the graph). Detection was performed by Phos-tag SDS-PAGE; ATPB was used as a loading control. Phosphatase-treated WT/*PMSK1-FLAG* and *phot*/*PMSK1-FLAG* samples were also loaded on the gels. A representative experiment from at least three independent experiments is shown. **d** Changes in the phosphorylation level of PMSK1-FLAG in WT/*PMSK1-FLAG*, WT/*PMSK1^S120A^-FLAG* and WT/*PMSK1^S120D^-FLAG* lines. Samples were collected after 24 h of acclimation to darkness (t = 0′) and 20′ after exposure to blue light (100 µmol photons m⁻² s⁻¹). Detection was performed by Phos-tag SDS-PAGE; ATPB was used as a loading control. Phosphatase-treated WT/*PMSK1-FLAG* sample was also loaded on the gels. "U" and "P" indicate the unphosphorylated and phosphorylated PMSK1-FLAG respectively. The blue or red bar to the left of the immunoblots indicates the light quality used. A representative experiment from at least three independent experiments is shown.

## PMSK1 acts downstream of PHOT to regulate starch metabolism in response to blue light

We applied CRISPR-CAS9 to disrupt *PMSK1* in the WT and in *phot*, thus generating the single *pmsk1* and the double *phot pmsk1* mutants (Supplementary Fig. 19). In comparison to WT, the *pmsk1* mutant accumulated 3-, 4- and 10-fold lower *GAP1* mRNA levels in white, blue and red light, respectively (Fig. 5a; when comparing *GAP1* mRNA levels at the middle of the day). Nevertheless, *GAP1* expression in *pmsk1* remained dependent on light-quality used, reaching higher expression levels under illumination with red light (Fig. 5a). However, this over-accumulation of *GAP1* mRNA in red light was not sufficient for over-accumulation of starch (Fig. 5b and Supplementary Fig. 20), as in the case of the *gap1 oe-3* line, which slightly overaccumulated *GAP1* mRNA but starch levels remained comparable to WT levels (Fig. 2h and i). As a result, we concluded that starch accumulation in *pmsk1* was light-quality-independent. The double mutant *phot pmsk1* accumulated 3-fold less *GAP1* mRNA and 2-fold less starch as compared to *phot* (Fig. 5a, b and Supplementary Fig. 20).

Taken together, our results show that PMSK1 plays a critical role in transducing the PHOT-mediated blue light signal to regulate starch metabolism in *Chlamydomonas*. Indeed, the *pmsk1* mutant has been instrumental in getting a clear picture of this distinct light-signalling pathway, summarized graphically in Fig. 6. Based on our data, we propose that PMSK1 is an activator of *GAP1* mRNA accumulation, and its

activity is regulated by the phosphorylation status of S120 in response to light quality. When PMSK1 is phosphorylated, it becomes active and promotes *GAP1* mRNA accumulation. Conversely, when it is not phosphorylated, PMSK1 is inactive and does not further promote *GAP1* mRNA accumulation. In addition to this regulation, GAP1 expression is influenced by other signals, possibly circadian, photosynthetic, or diurnal rhythms, responsible for the expression pattern observed over the course of the day. Red light drives the phosphorylated form of PMSK1 (PMSK1-P; Fig. 3c), resulting in high *GAP1* mRNA accumulation (Fig. 4a). However, in white and blue light, dephosphorylation of active PMSK1-P results in an accumulation of un-phosphorylated PMSK1 (PMSK1-U; Fig. 3c), reducing the relative abundance of PMSK1-P within the total PMSK1 pool. This shift toward PMSK1-U decreases the proportion of PMSK1-P, leading to reduced GAP1 mRNA levels (Fig. 4a). In the *pmsk1* mutant, *GAP1* mRNA levels (Fig. 5a) are similar to those in WT/PMSK1-S120A (unphosphorylated form; Fig. 4a), showing lower expression at 12 h. However, there is still regulation in red light, which is abolished in the *phot pmsk1* double mutant (Fig. 5a), suggesting the existence of other regulators of *GAP1* that are controlled by blue light independently of PMSK1, although PMSK1 appears to play a major role.

Our model (Fig. 6), which suggests that PMSK1-P acts as an activator of *GAP1* mRNA accumulation, explains why the PMSK1 mutation has the greatest impact on *GAP1* mRNA levels in red light (Fig. 5a). In red light, PMSK1-P is most abundant in WT, making the mutation's

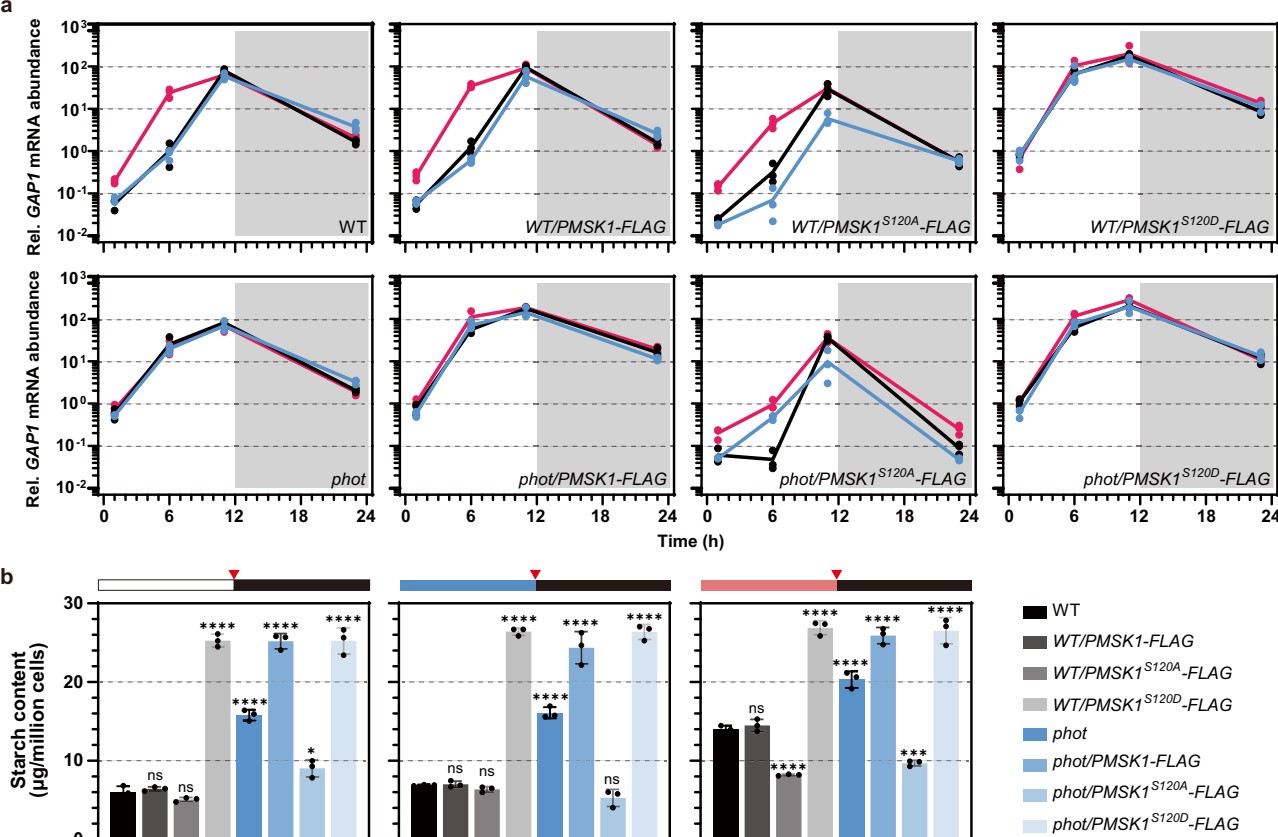

**Fig. 4 | PMSK1 phosphorylation status controls starch metabolism in *Chlamydomonas reinhardtii*.** *GAP1* transcription level (**a**) and starch content (**b**) of various PMSK1 overexpression lines synchronized to a 12/12 light dark cycle under different light qualities. Data are presented as mean ± SD (n = 3 biologically independent samples). Phases are indicated by white and gray shading in (**a**). Line colors indicated light qualities. Black, white light; Red, red light; Blue, blue light. In **b** red triangles indicate sample collection time. The dark phase is indicated by black bars above the graphs; the light phase by white, blue, or red bars, depending on the light quality used. The statistical significance was determined using two-way ANOVA with Dunnett's multiple comparisons tests (**a**) of log10 transformed mRNA data as indicated in the graphs. The statistical significance was determined using one-way ANOVA with Dunnett's multiple comparisons tests (**b**). Asterisks indicated the p-values compare to WT (*p < 0.05; ***p < 0.001; ****p < 0.0001; ns, not significant). In some cases, the error bars are smaller than the data point symbols. Detailed statistical analyses are presented in the Source Data File.

effect more pronounced. In contrast, in white or blue light, PHOT promotes PMSK1 dephosphorylation, reducing PMSK1-P levels and diminishing the mutation's impact in these conditions.

## Discussion

In this work we report the discovery of an unrecognized light-signalling pathway linking blue-light perception by PHOT with starch accumulation in *Chlamydomonas*. Our data solve a four-decades-long question about the reasons for starch accumulation in green algae predominantly under red light[31]. We showed that PHOT regulates the phosphorylation of a specific serine residue (S120) on a yet uncharacterized kinase, PMSK1. In its phosphorylated state, PMSK1 transduces a signal that controls the accumulation of *GAP1* mRNA (Fig. 6). Since GAP1 is involved in the generation of phosphorylated sugars, as precursors of starch synthesis, our findings demonstrate how this light-signalling pathway leads to improvement of starch metabolism.

Our work establishes PMSK1 as a kinase involved in PHOT-dependent signaling in algae, highlighting its distinct role in this pathway. Yet, PHOT may not be the only blue-light responsive protein suppressing starch accumulation in *Chlamydomonas*; as our data show, application of red light in the *phot* mutant results in higher accumulation of starch compared to white or blue illuminated cells (Fig. 1d), suggesting the presence of additional blue-light responsive protein(s) repressing starch accumulation, as illustrated in our model (Fig. 6). How PHOT, a kinase, is involved in the dephosphorylation of PMSK1 is an

intriguing question that needs further investigation. One of the possibilities is that a protein phosphatase (indicated PPase in Fig. 6) is a missing component in our proposed model; PHOT may mediate the activity of this PPase to dephosphorylate S120 of PMSK1. In *Arabidopsis*, which encodes two phototropins, Phot1 and Phot2, blue light induces the phosphorylation of NPH3 (NON-PHOTOTROPIC HYPOCOTYL3) at serine 744 (S744) in a Phot1-dependent manner. This phosphorylation creates a 14-3-3 binding site enabling NPH3 to associate with 14-3-3 proteins. Subsequently NPH3 gets dephosphorylated[32,33]. Whether *Chlamydomonas* PHOT operates via a similar mechanism—interacting with and phosphorylating PMSK1 prior to its dephosphorylation at S120—remains to be determined.

PMSK1 is found to belong to a family of Serine/threonine-protein kinases (named PMSK-like family, see material and methods) which is conserved in green algae and vascular plants. When searching for PMSK1-like sequences in the *Arabidopsis* genome in the NCBI database[34], HIGH LEAF TEMPERATURE 1 (HT1)[35] and CONVERGENCE OF BLUE LIGHT AND CO2 1/2 (CBC1/2)[36] are found in addition to the PMSK-like Arabidopsis members (Supplementary Figs. 21–24 and Supplementary Text), proteins that have been shown to mediate responses to $CO_2$ and blue light in *Arabidopsis*. Yet, while *Arabidopsis* CBC1/2/HT1 acts to stimulate stomatal opening by inhibiting S-type anion channels[36], our findings demonstrate that the regulatory function of its *Chlamydomonas* counterpart PMSK1 controls starch metabolism through the transcriptional regulation of *GAP1*.

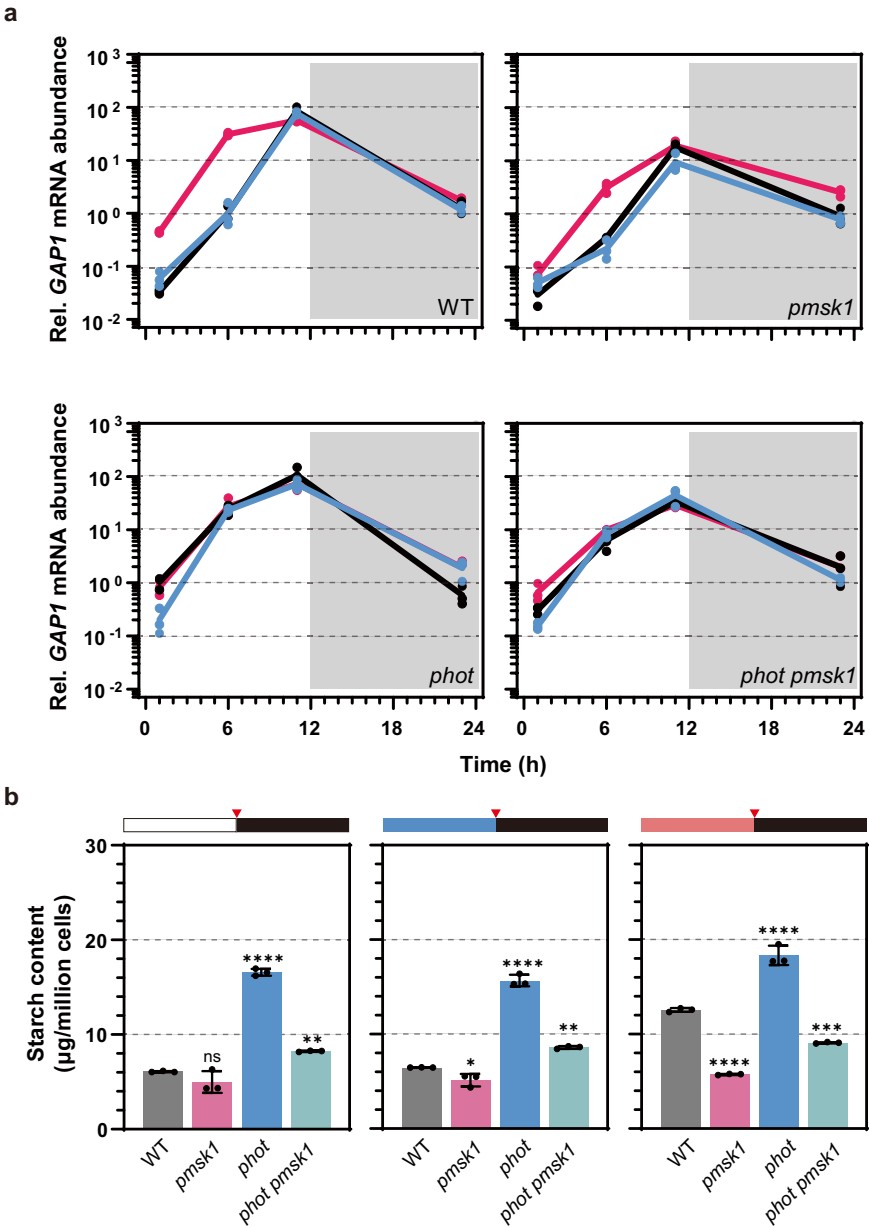

**Fig. 5 | PHOT regulates starch metabolism via PMSK1 and GAP1 in *Chlamydomonas reinhardtii*. a** *GAP1* transcription level and **b** starch content in single and double *phot* and *pmsk1* mutants synchronized to a 12/12 light dark cycle under different light qualities. Data are represented as mean ± SD (n = 3 biologically independent samples). Line colors indicated light qualities. Black, white light; Red, red light; Blue, blue light. Red triangles indicate sample collection time. Dark phase is indicated by black bars above the graphs; the light phase by white, blue, or red bars, depending on the light quality used. Red triangles indicated samples collection time. The statistical significance was determined using two-way ANOVA with Dunnett's multiple comparisons tests (**a**) of log10 transformed mRNA data as indicated in the graphs. The statistical significance was determined using one-way ANOVA with Dunnett's multiple comparisons tests (**b**). Asterisks indicated the p-values compare to WT (*p < 0.05; **p < 0.01; ***p < 0.001; ****p < 0.0001; ns, not significant). In some cases, the error bars are smaller than the data point symbols. Detailed statistical analyses are presented in the Source Data File.

In *Arabidopsis*, phototropins (Phot1 and Phot2) play a crucial role in regulating starch degradation in guard cells in response to blue light, a process essential for stomatal opening, which manages gas exchange and water balance in plants. Upon exposure to blue light, phototropin signaling triggers a pathway that leads to the rapid breakdown of starch in guard cells. This involves the coordinated action of the enzymes β-amylase 1 (BAM1) and α-amylase 3 (AMY3). The starch degradation is linked to the activation of the plasma membrane H +-ATPase, which promotes stomatal opening and contributes to overall plant growth[20]. Our findings reveal yet another fascinating difference between vascular plants and *Chlamydomonas* phototropins; in contrast to the Arabidopsis Phots, Chlamydomonas PHOT acts on the mRNA level of a key metabolic enzyme GAP1 to modulate cellular accumulation of starch.

In the green microalgae *Chlorella*, a possible explanation for why blue light represses starch accumulation[22] may lie in early studies showing that under blue light carbohydrate catabolism is enhanced[37,38]. While the specific role of GAP1 in *Chlorella* remains to be investigated, we recently demonstrated that under red light, *Chlorella* accumulated significant starch levels even after a prolonged 7-day exposure[39], in accordance to earlier short-term experiments[22]. In *Chlamydomonas*, our results showed that both starch synthesis- and catabolism-related genes were upregulated in the *phot* mutant (Supplementary Fig. 9). Although proteomics does not necessarily capture

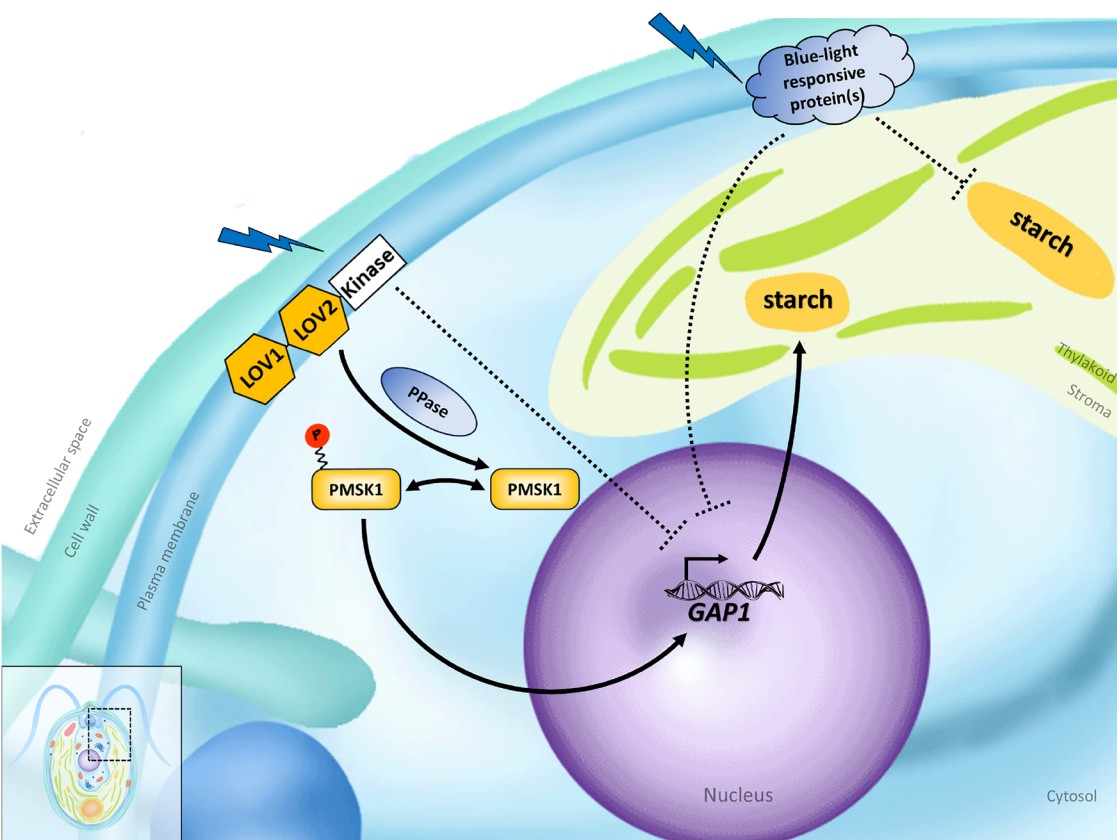

**Fig. 6 | Proposed model depicting the mechanisms of blue-light dependent regulation of starch metabolism in *Chlamydomonas reinhardtii*.** Phosphorylated PMSK1 at S120 (PMSK1-P) activates GAP1, leading to increased starch biosynthesis. In the presence of blue light, PHOTOTROPIN detects the signal and initiates the dephosphorylation of PMSK1, resulting in decreased GAP1 mRNA levels and reduced starch accumulation. PPase is a hypothetical phosphatase that could dephosphorylate PMSK1-P in a PHOTOTROPIN-dependent manner. The model also includes a PHOTOTROPIN-dependent but PMSK1-independent repression of GAP1, as GAP1 in *pmsk1* mutants remains regulated by red light—a regulation lost in the *phot pmsk1* double mutant (Fig. 5a). Additionally, other blue-light-responsive proteins are proposed to independently inhibit starch accumulation or GAP1 outside of the PHOT and GAP1 pathways. This is based on observations that: (i) *phot* mutants accumulate more starch under red light (Fig. 1d), despite consistently high GAP1 mRNA levels across light conditions (Fig. 2d), and (ii) red light enhances GAP1 mRNA in the *phot-kin* strain (Fig. 2g).

enzyme activity, it is worth mentioning that our whole-cell proteomics data (Supplementary Data 2) did not reveal any statistically significant changes in the levels of alpha-amylase, the enzyme responsible for the major hydrolytic activity involved in starch degradation in *Chlamydomonas*[40].

In summary, our findings not only demonstrate that the PHOT-PMSK1 pathway regulates starch metabolism in *Chlamydomonas*, but they also shed light on the importance of this mechanism within the context of the organism's diurnal rhythm. The regulatory effect of PHOT-PMSK1 plays a crucial role in modulating GAP1 mRNA levels, particularly at the onset of illumination following the dark phase. By acting through blue light, PHOT-PMSK1 fine-tunes the induction of GAP1, preventing an immediate spike in its expression. This controlled response ensures that starch synthesis does not overwhelm the cell's energy reserves, allowing for a balanced distribution of resources between starch production and other essential cellular functions. This regulation likely prevents excessive starch accumulation, which could otherwise compromise cellular health, highlighting the critical nature of this finely tuned-mechanism.

Nonetheless, the higher starch accumulation in the *phot* mutant under the low light intensity conditions used in our experiments, does not affect growth and photosynthesis (Supplementary Fig. 5). This finding indicates that this PHOT-mediated light-signaling pathway overcomes a key resource allocation trade-off present in *Chlamydomonas*, whereby carbon, fixed by photosynthesis, is allocated to energy reserves (e.g. starch) at the cost of growth[14]. Our findings pave

the way for the application of precise kinase engineering of PMSK1 as a sustainable way to produce starch from green microalgae in biotechnological applications.

## Methods

### Statistics
Statistical methods were not used to predetermine the sample size. The experiments were not randomized, and the investigators were not blinded to allocation during experimental procedures and data assessment.

### Algal Material
The strains used in this study included *Chlamydomonas phot* (defective in *PHOT*; gene ID: Cre03.g199000) and *phot-C1* (*phot* strain complemented with WT *PHOT* gene), as well as their background strain CC-125, which have been previously described[25]. Additionally, *Chlamydomonas acry*[24] (defective in animal-type cryptochrome, aka *aCRY*; gene ID: Cre06.g278251), *pcry* (defective in plant-type cryptochrome, aka *pCRY*; gene ID: Cre06.g295200), and *acrypcry* (defective in both animal-type and plant-type cryptochrome) were generated through CRISPR-CAS9 provided by following the protocol described in ref. 24. The *pmsk1* (defective in phototropin-mediated signaling kinase 1, aka *PMSK1*; gene ID: Cre16.g659400) *and photpmsk1* (defective in both *PHOT* and *PMSK1*) mutants were generated using insertional CRISPR-Cas9 RNP method described by Kim et al.[41] with a few modifications. The target sgRNA sequence of *PMSK1* was

designed by Cas-Designer (http://www.rgenome.net/cas-designer) and selected considering the recommendation guideline. To induce early termination of translation, the sgRNA targets were selected in exon 2 (Supplementary Data 1). To form an RNP complex in vitro, 100 μg of purified Cas9 protein (Cas9 expression plasmid: Plasmid #62934, addgene, US) and 70 μg of sgRNA synthesized by using GeneArt™ Precision gRNA Synthesis Kit (ThermoFisher, US), were mixed gently. For efficient and fast screening, 0.5 μg of paromomycin-resistance gene cassette was co-transformed with RNP complex. The *Chlamydomonas* cell wall was permeabilized by treatment of Max Efficiency buffer (ThermoFisher, US) following the manufacturer's protocol. The *Chlamydomonas* transformation was performed in the 4 mm gap electroporating cuvette by electroporation with the specific parameter (600 V, 50 μF, 200 Ω). One day after transformation, cells were plated on TAP medium containing 1.5% agar and paromomycin (25 μg/ml). Once colonies appear after transformation, genomic DNA PCR and Sanger sequencing were performed to validate knockout events.

To prepare transgenic lines with a knockdown of GAP1, pChlamiRNA3int-GAP1 was transformed into the *phot* strain. Generation of amiRNA plasmids was performed according to[42]. The oligonucleotides designed for targeting GAP1 (Data S1) using the WEB MicroRNA Designer platform (WMD3: http://wmd3.weigelworld.org/cgi-bin/webapp.cgi. Ossowski Stephan, Fitz Joffrey, Schwab Rebecca, Riester Markus and Weigel Detlef, personal communication) were annealed and ligated into pChlamiRNA3int (SpeI digested) to create pChlamiRNA3int-gap1. Transformed cells were selected and further checked by RT-qPCR.

For the preparation of transgenic lines overexpressing GAP1(Glyceraldehyde 3-phosphate dehydrogenase, aka *GAP1*; gene ID: Cre12.g485150) or different versions of PMSK1, the genomic sequences of *GAP1* and *PMSK1* were PCR amplified from genomic DNA of *Chlamydomonas* CC-125 and cloned into pLM005 in-frame with a C-terminal Venus-3Xflag using Gibson Assembly[43], and then transformed into WT or *phot* strains. The primers used for different gene amplification and point mutations are described in Supplementary Data 1. All *Chlamydomonas reinhardtii* strains used in this study are listed in Supplementary Data 4.

### *Chlamydomonas reinhardtii* cultivation
All *Chlamydomonas* strains were maintained on solid Tris-acetate-phosphate (TAP)[44] agar plates with or without appropriate antibiotic at 22 °C and 5 μmol photons m$^{-2}$ s$^{-1}$. Prior to the start of the experiments, cells were cultured in 50 mL TAP medium in 250 ml Erlenmeyer flasks at 23 °C, 120 rpm/min and 15 μmol photons m$^{-2}$ s$^{-1}$. The experiments were conducted in Sueoka's high salt medium (HSM)[45] at an initial cell density of 1 million cells/ml at 50 μmol photons m$^{-2}$ s$^{-1}$ unless otherwise stated.

For continuous light experiments, the cells were transferred to HSM medium and grew at 23 °C, 120 rpm/min, and 50 μmol photons m$^{-2}$ s$^{-1}$.

For synchronized experiments, cells were grown in HSM for at least 5 days under a 12 h light/12 h dark cycle under white light or different light qualities (light intensity was set at 50 μmol photons m$^{-2}$ s$^{-1}$; temperature was 18 °C in the dark and 23 °C in the light). The light spectrum of the LED lighting system used in this study is the same as previously described[46].

### Transformation of *Chlamydomonas reinhardtii*
The transformation was performed by electroporation, which follows the protocol of Zhang et al.[47] with minor modification. Cells for transformation were collected at 1–2 h before the end of the light phase in a synchronized (12 h light/12 h dark) culture. For three reactions 11 ng/kb linearized plasmid was mixed with 400 μl of $1.0 \times 10^7$ *Chlamydomonas reinhardtii* cells/ml and electroporated at a volume of

125 ml in a 2-mm-gap electro cuvette using a NEPA21 square-pulse electroporator, using two poring pulses of 250 and 150 V for 8 ms each, and five transfer pulses of 50 ms each starting at 20 V with a "decay rate" of 40% (i.e., successive pulses of 20, 12, 7.2, 4.3, and 2.6 V). Electroporated cells were immediately transferred to a 15 ml centrifugation tube containing 9 ml TAP plus 40 mM sucrose. After overnight dark incubation, cells were collected by centrifugation and spread on TAP agar plates which contain the appropriate antibiotic (20 μg/ml paromomycin or 7.5 μg/ml zeocin or 20 μg/ml hygromycin B). Transformants typically appear after 5–7 days.

The putative antibiotic-resistant transformants were transferred into individual wells of a 96-well, flat-bottom transparent microplate, with each well containing 250 μl of TAP medium. Cultures were grown for 3 days under 15 μmol photons m$^{-2}$ s$^{-1}$ light without shaking, refreshed by replacing half of the culture with fresh medium, and allowed to grow for an additional day. Transformants were screened for Venus expression using a fluorescent microplate reader (Tecan Group Ltd, Switzerland), with parameters including Venus (excitation 515/12 nm and emission 550/12 nm) and chlorophyll (excitation 440/9 nm and emission 680/20 nm). The fluorescence signal was normalized to the chlorophyll fluorescence signal, and colonies with a high Venus/chlorophyll value were selected as putative complemented strains. These putative positive transformants were further validated by western blotting and RT-qPCR.

### Cell counting
Cell concentration was determined using either a hemocytometer or an automated cell counter (Countess II FL, Thermo Fisher, US). For the hemocytometer method, a 10 μL aliquot was mixed with 5% acetate and then loaded onto the hemocytometer, where cells were counted manually under a light microscope in four 1 mm$^2$ squares. For automated counting, a 10 μL aliquot was similarly mixed with 5% acetate and loaded into a disposable chamber for analysis using the automated cell counter. Both methods were performed in triplicate, and the mean values were subsequently reported.

### RNA extractions and RT-qPCR analysis
Total RNA for RNA-seq and RT-qPCR was extracted using RNeasy Mini Kit (Qiagen, Germany) and treated with the RNase-Free DNase Set (Qiagen, Germany). One microgram total RNA was reverse transcribed with oligo dT using Sensifast cDNA Synthesis kit (Meridian Bioscience, US). qPCR reactions were performed and quantitated in a Bio-Rad CFX96 system using SsoAdvanced Universal SYBR Green Supermix (BioRad, US). The *CBLP* gene[48] served as the housekeeping control and relative fold differences were calculated on the basis of the $\Delta C_t$ method ($2^{-(Ct\,target\,gene\,-\,Ct\,CBPL)}$)[49–51]. All primers used for the RT-qPCR analyses were synthesized by ThermoFisher (US) or IDT (Integrated DNA Technologies, Inc. Coralville, Iowa, US) and were presented in Supplementary Data 1.

### Analyses of total starch content
The total starch content of samples collected daily was determined using Total Starch Assay Kit (K-TSTA-100A, Megazyme, Ireland) as described in its instruction with modifications. The results were calculated according to the standard curve made with glucose solution after starch digestion. To prepare the samples for glucose determination, 10 mL of the liquid culture was pelleted by centrifugation and resuspended in 40 μL of 80% (v/v) ethanol. Next, 400 μL of cold 1.7 M sodium hydroxide solution was added, and the samples were incubated on ice for 15 min. Following this, 1.6 mL of sodium acetate buffer containing calcium chloride (5 mM) was added and mixed well. Subsequently, 20 μL of α-amylase and 20 μL of amyloglucosidase were added, and the samples were incubated at 50 °C for 30 min. The supernatant was collected by centrifugation at 23,000 × g for 5 min and will be ready for glucose determination.

## Analyses of Total Protein and Lipid content

The total protein content of samples collected daily was determined using BCA Protein Assay Kit (ThermoFisher, US) with the standardized protocol. The total lipid content of samples collected daily was determined using sulfo-phospho-vanillin (SPV) method[52].

## Transmission electron microscopy

Cells were harvested by centrifugation at $700 \times g$ for 5 min, washed two times in 0.1 M PB (phosphate buffer, pH 7.4), and then were fixed in 0.1 M PB containing 2.5% (v/v) glutaraldehyde for 2 h at room temperature and stored overnight at 4 °C. The cells were then washed five times in 0.1 M PB before being fixed by a 1 h incubation on ice in 0.1 M PB containing 2% osmium and 1.5% ferricyanide potassium. After being washed five times with 0.1 M PB, the samples were resuspended in 0.1 M PB containing 0.1% (v/v) tannic acid and incubated for 30 min in the dark at room temperature. The cells were washed five times with 0.1 M PB, dehydrated in ascending sequences of ethanol, infiltrated with an ethanol/Epon resin mixture, and finally embedded in Epon. Ultrathin sections (50–70 nm) were prepared with a diamond knife on a PowerTome ultramicrotome (RMC Boeckeler, US) and collected on 200 μm nickel grids. The ultrathin sections were examined on a Philips CM120 transmission electron microscope operating at 80 kV.

## Confocal microscopy

The preparation of samples for confocal microscopy followed the protocol reported by Mackinder et al.[53]. The confocal microscope used in the study was from the cell-imaging platform at IBS, Grenoble, France. All confocal microscopy images were analyzed using Fiji[54].

## Immunoblotting

Protein samples of whole cell extracts (5 μg protein) were loaded on 4–20% SDS-PAGE gels (Mini-PROTEAN TGX Precast Protein Gels, Bio-Rad, US) and blotted onto nitrocellulose membranes. Antisera against ATPB (AS05085, 1:10,000, https://www.agrisera.com/en/artiklar/atpb-beta-subunit-of-atp-synthase.html) and FLAG (AS20 4442, 1:5000, https://www.agrisera.com/en/artiklar/dykdddk-binds-to-sigma-flag-polyclonal.html) were from Agrisera (Sweden); antiserum PHOT (LOV1 domain, 1:5000) was previously described (Fig. 2g in ref. 55). ATPB was used as a loading control. The anti-rabbit horseradish peroxidase–conjugated antiserum (Jackson Immuno Research, US) was used for detection at 1:10,000 dilution. The blots were developed with ECL detection reagent, and images of the blots were obtained using ImageQuant 800 (Cytiva, UK). For the densitometric quantification, data were normalized with ATPB.

## Phos-tag gel electrophoresis

Double-layer Phos-tag gels with a concentration of 12% (w/v) acrylamide/bisacrylamide 37.5:1 and 65 mM of Phos-Tag (Wako Pure, US) were prepared as in ref. 56, with the exception that $Zn(NO_3)^2$ was added equimolarly to the samples to compensate for the absence of EDTA in the lysis buffer. The gels were denatured for 30 min at 37 °C prior to loading. In vitro dephosphorylation involved resuspending a cell pellet in 5 mM of HEPES at pH 7.5, 10 mM of EDTA, and 1% (v/v) TritonX 100. An aliquot containing 10 mg of protein was then subjected to lambda protein phosphatase reaction mix following the manufacturer's instructions (New England Biolabs, US) for 1 h at 30 °C, in accordance to ref. 13.

## Fluorescence-based measurements

Fluorescence-based photosynthetic parameters were measured with a pulse-modulated amplitude fluorimeter (MAXI-IMAGING-PAM, Heinz-Waltz GmbH, Germany). Prior to the onset of the measurements, cells were acclimated to darkness for 15 min. Chlorophyll fluorescence was recorded under different intensities of actinic light; starting with measurements in the dark (indicated as D below the x-axis of the graphs), followed by measurements at 21 μmol photons $m^{-2}$ $s^{-1}$ (indicated as L1 below the x-axis of the graphs) and 336 μmol photons $m^{-2}$ $s^{-1}$ (indicated as L2 below the x-axis of the graphs) and finishing with measurements of fluorescence relaxation in the dark. The effective photochemical quantum yield of photosystem II was calculated as $Y(II) = (Fm' - F)/Fm'$; $F$ and $Fm'$ are the fluorescence yield in steady state light and after a saturating pulse in the actinic light, respectively.

## Phosphoproteomics analysis

Protein extraction: *C. reinhardtii* pellets were resuspended in 2000 μL of lysis buffer (100 mM Tris-HCl, PhosphoSTOP inhibitors, protease inhibitors) and ultrasonicated in the Covaris for 4 min each. Samples were diluted by adding 2000 μL of dilution buffer (100 mM Tris-HCl, 5 mM TCEP, 30 mM chloroacetamide, 1 mM sodium orthovanadate, phosphoSTOP inhibitors, 1 mM magnesium chloride) and 1 μL Benzonase. Lysates were shaken at 25 °C for 1 h. 8 mL of methanol was added to each sample, followed by 3 mL chloroform and 3 mL water, with vortexing after each subsequent addition. Samples were centrifuged for 10 min at $3220 \times g$ and the top layer was removed, leaving the interphase intact. An additional 10 mL of methanol was added and the samples were centrifuged for 20 min at $3220 \times g$ and the supernatant was removed. Protein pellets were allowed to dry at RT and resuspended in 1 mL of digestion buffer (100 mM Tris-HCl, 2 M urea). Proteins were digested with 12 μg of trypsin overnight and cleaned up via C18 SPE cartridges. Samples were resuspended in 250 μL of water and a BCA assay was performed to determine peptide concentration. Ten micrograms of digested protein was taken for global analysis, and 500 μg was used for phosphopeptide enrichment.

Phosphopeptide enrichment: Phosphopeptides were enriched using a ProPac Fe-IMAC column (ThermoFisher, US) on a Shimadzu Prominence HPLC system. Before enrichment, the column was charged with 25 mM $FeCl_3$ in 100 mM acetic acid. Mobile phase A consisted of 30% acetonitrile in water (v/v) with 0.07% trifluoroacetic acid (v/v). Mobile phase B consisted of 0.3% ammonium hydroxide in water (v/v). Tryptic peptides were diluted to 30% acetonitrile and injected on the column at a flow rate of 0.2 ml/min. After 3 min of loading, flow rate was increased to 2 mL/min. Peptides were eluted by rapidly ramping the gradient to 50% B. Fractions containing phosphopeptides were cleaned up with C18 SPE cartridges (Waters, US) and resuspended in 20 μL of LC-MS water prior to mass spectrometry analysis.

LC-MS analysis: Phosphopeptide samples were analyzed using a nanoACQUITY UPLC (Waters, US) coupled to a TripleTOF 5600 mass spectrometer (Sciex, Canada). Mobile phase A consisted of water with 0.1% formic acid and mobile phase B was acetonitrile with 0.1% formic acid. Injections were made to a Symmetry C18 trap column (100 Å, 5 μm, 180 μm × 20 mm; Waters, US) with a flow rate of 5 μL/min for 3 min using 99% A and 1% B. Peptides were then separated on an HSS T3 C18 column (100 Å, 1.8 μm, 75 μm × 250 mm; Waters, US) using a linear gradient of increasing mobile phase B at a flow rate of 300 nL/min. Mobile phase B increased from 5% to 40% in 90 min before ramping to 85% in 5 min, where it was held for 5 min before returning to 5% in 2 min and re-equilibrating for 13 min.

The mass spectrometer was operated in positive polarity mode. MS survey scans were accumulated across an $m/z$ range of 350–1600 in 250 ms optimized at ≥30,000 resolution. For data-dependent acquisition, the mass spectrometer was set to automatically switch between MS and MS/MS experiments for the first 20 features above 150 counts having +2 to +5 charge state. Precursor ions were fragmented using rolling collision energy and accumulated in high sensitivity mode for 85 ms across an $m/z$ range of 100–1800 optimized at ≥30,000 resolution. Dynamic exclusion for precursor m/z was set to 8 s.

Bioinformatic analysis: Raw data files were imported into Progenesis for peak alignment and quantification. Spectra were searched in Mascot against the *C. reinhardtii* phytozome database (v6.1) using a precursor/fragment tolerance of 15 ppm/0.1 Da, trypsin specificity, two

possible missed cleavages, fixed modification cysteine carbamido-methylation, and variable modifications of methionine oxidation, protein N-term acetylation, and phosphorylation (STY). Identifications were imported back into Progenesis for peak assignment, and statistical analysis was performed using the QuantifyR workflow, which can be found on the Hicks Lab Github (github.com/hickslab/QuantifyR). Phosphoproteomics data have been deposited to the ProteomeXchange Consortium via the PRIDE partner repository[57] with the dataset identifier PXD045599.

## Lipidomics

Glycerolipids were extracted from freeze-dried cell pellets frozen immediately in liquid nitrogen after harvesting. Once freeze-dried, cell pellets were resuspended in 4 mL of boiling ethanol for 5 min to prevent lipid degradation and lipids were extracted according to[58] by addition of 2 mL methanol and 8 mL chloroform at room temperature. The mixture was then saturated with argon and stirred for 1 h at room temperature. After filtration through glass wool, cell remains were rinsed with 3 mL chloroform/methanol 2:1, v/v, and 5 mL of NaCl 1% were then added to the filtrate to initiate biphase formation. The chloroform phase was dried under argon before solubilizing the lipid extract in pure chloroform. Total glycerolipids were quantified from their fatty acids: in an aliquot fraction, a known quantity of 15:0 was added and the fatty acids present were transformed as methyl esters (FAME) by a 1-h incubation in 3 mL 2.5% $H_2SO_4$ in pure methanol at 100 °C[59]. The reaction was stopped by addition of 3 mL water and 3 mL hexane. The hexane phase was analyzed by gas chromatography-flame ionization detector (GC-FID) (Perkin Elmer, US) on a BPX70 (SGE; Trajan Scientific and Medical location, Australia) column. FAME were identified by comparison of their retention times with those of standards (Sigma, US) and quantified by the surface peak method using 15:0 for calibration.

The lipid extracts corresponding to 25 nmol of total fatty acids were dissolved in 100 µL of chloroform/methanol [2/1, (v/v)] containing 125 pmol of each internal standard. Internal standards used were PE 18:0-18:0 and DAG 18:0-22:6 from Avanti Polar Lipid and SQDG 16:0–18:0 extracted from spinach thylakoid[60] and hydrogenated as described in ref. 61. Lipids were then separated by HPLC and quantified by MS/MS.

The HPLC separation method was adapted from ref. 62. Lipid classes were separated using an Agilent 1200 HPLC system using a 150 × 3 mm (length × internal diameter) 5 µm diol column (Macherey-Nagel, Germany), at 40 °C. The mobile phases consisted of hexane/isopropanol/water/ammonium acetate 1 M, pH5.3 [625/350/24/1, (v/v/v/v)] (A) and isopropanol/water/ammonium acetate 1 M, pH5.3 [850/149/1, (v/v/v)] (B). The injection volume was 20 µL. After 5 min, the percentage of B was increased linearly from 0% to 100% in 30 min and stayed at 100% for 15 min. This elution sequence was followed by a return to 100% A in 5 min and an equilibration for 20 min with 100% A before the next injection, leading to a total runtime of 70 min. The flow rate of the mobile phase was 200 µL/min. The distinct glycerophospholipid classes were eluted successively as a function of the polar head group.

Mass spectrometric analysis was done on a 6470 triple quadrupole mass spectrometer (Agilent, US) equipped with a Jet stream electrospray ion source under following settings: Drying gas heater: 230 °C, Drying gas flow 10 L/min, Sheath gas heater: 200 °C, Sheath gas flow: 10 L/min, Nebulizer pressure: 25 psi, Capillary voltage: ±4000 V, Nozzle voltage ±2000. Nitrogen was used as collision gas. The quadrupoles Q1 and Q3 were operated at widest and unit resolution respectively. DGTS analysis was carried out in positive ion mode by scanning for precursors of m/z 236 at a collision energy (CE) of 55 eV. SQDG analysis was carried out in negative ion mode by scanning for precursors of m/z −225 at a CE of −55 eV. PE, PI, PG, MGDG, and DGDG measurements were performed in positive ion mode by scanning for neutral losses of 141 Da, 277 Da, 189 Da, 179 Da, and 341 Da at CEs of 29 eV, 21 eV, 25 eV, 8 eV, and 11 eV, respectively. Quantification was

done by multiple reaction monitoring (MRM) with 30 ms dwell time. DAG and TAG species were identified and quantified by MRM as singly charged ions $[M + NH_4]^+$ at a CE of 19 and 26 eV respectively with 30 ms dwell time. Mass spectra were processed by MassHunter Workstation software (Agilent, US) for identification and quantification of lipids. Lipid amounts (pmol) were corrected for response differences between internal standards and endogenous lipids and by comparison with a quality control (QC). QC extract correspond to a known lipid extract from *Chlamydomonas* cell culture qualified and quantified by TLC and GC-FID as described in ref. 63.

## Proteomics analysis

Proteins from total extracts of three biological replicates of WT and *phot Chlamydomonas* reinhardtii were solubilized in Laemmli buffer and heated for 10 min at 95 °C. They were then stacked in the top of a 4–12% NuPAGE gel (ThermoFisher, US), stained with Coomassie blue R-250 (Bio-Rad, US) before in-gel digestion using modified trypsin (Promega, US) as previously described[64]. The resulting peptides were analyzed by online nanoliquid chromatography coupled to MS/MS (Ultimate 3000 RSLCnano and Q-Exactive HF, ThermoFisher, US) using a 180-min gradient. For this purpose, the peptides were sampled on a precolumn (300 µm × 5 mm PepMap C18, ThermoFisher, US) and separated in a 75 µm × 250 mm C18 column (Reprosil-Pur 120 C18-AQ, 1.9 µm, Dr. Maisch, Germany). The MS and MS/MS data were acquired using Xcalibur (V2.8, ThermoFisher, US).

Peptides and proteins were identified by Mascot (V2.8.0, Matrix Science) through concomitant searches against the *C. reinhardtii* phytozome database (V5.6) (19526 sequences), the mitochondrion and chloroplast protein sequences (downloaded from NCBI, respectively 69 and 8 proteins), and a homemade database containing the sequences of classical contaminant proteins found in proteomic analyses (e.g. human keratins, trypsin). Trypsin/P was chosen as the enzyme and two missed cleavages were allowed. Precursor and fragment mass error tolerances were set at respectively at 10 and 20 ppm. Peptide modifications allowed during the search were: Carbamidomethyl (C, fixed), Acetyl (Protein N-term, variable) and Oxidation (M, variable). The Proline software[65] (V2.2.0) was used for the compilation, grouping, and filtering of the results (conservation of rank 1 peptides, peptide length ≥6 amino acids, false discovery rate of peptide-spectrum-match identifications <1%[66], and minimum of one specific peptide per identified protein group). MS data have been deposited to the ProteomeXchange Consortium via the PRIDE partner repository[67] with the dataset identifier PXD046943. Proline was then used to perform a MS1 label-free quantification of the identified protein groups based on razor and specific peptides.

Statistical analysis was performed using the ProStaR software[68] based on the quantitative data obtained with the three biological replicates analyzed per condition. Proteins identified in the contaminant database, proteins identified by MS/MS in less than two replicates of one condition, and proteins quantified in less than three replicates of one condition were discarded. After log2 transformation, abundance values were normalized using the variance stabilizing normalization (vsn) method, before missing value imputation (SLSA algorithm for partially observed values in the condition and DetQuantile algorithm for totally absent values in the condition). Statistical testing was conducted with limma, whereby differentially expressed proteins were selected using a log2(Fold Change) cut-off of 1 and a p-value cut-off of 0.00912, allowing to reach a false discovery rate inferior to 1% according to the Benjamini-Hochberg estimator. Proteins found differentially abundant but identified by MS/MS in less than two replicates, and detected in less than four replicates, in the condition in which they were found to be more abundant were invalidated (p-value = 1).

## GO enrichment analysis

GO term enrichment was tested for all proteins significantly differential abundant at a false discovery rate below 5% using the Benjamini-

Hochberg estimator[69]. GO annotation of proteins was obtained from phytozome database (v.5.6) and all ancestral GO terms were added to a protein using the R package GO.db. P-values were obtained according to the null hypothesis, that the number of differential abundance proteins bearing a GO term is a random variable whose probability distribution is described by the hypergeometric distribution. The false discovery rate was controlled below 5% using the Benjamini-Hochberg-estimator[69]. Only GO terms linked to at least 6 measured proteins were tested.

### Statistical analysis

Prism (GraphPad Software) was used for statistical analysis and all error bars represent standard deviation. ANOVA tests and t-tests were performed, with the p-values or degree of significance provided in the figures and the legends.

### Reporting summary

Further information on research design is available in the Nature Portfolio Reporting Summary linked to this article.

### Data availability

All biological material described in this study is available upon request. All data are available in the main text or the supplementary materials. Raw values and statistical analyses of figures can be found in the Source Data File. Raw data for lipidomic analysis can be found in the Source Data file. Global MS-based proteomic data have been deposited to the ProteomeXchange Consortium via the PRIDE partner repository[57] with the dataset identifier PXD046943. Phosphoproteomics data have been deposited to the ProteomeXchange Consortium via the PRIDE partner repository[57] with the dataset identifier PXD045599. Source data are provided with this paper.

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

## Acknowledgements

This work used the platforms of the Grenoble Instruct Centre (ISBG; UMS 3518 CNRS-CEA-UJF-EMBL) with support from FRISBI (ANR-10-INSB-05-02) and GRAL (ANR-10-LABX-49-01) within the Grenoble Partnership for Structural Biology (PSB). We would like to thank the following agencies for funding: The Human Frontiers Science Program through the funding of the project RGP0046/2018 (D.P., Z.N.); the French National Research Agency in the framework of the Young Investigators program ANR-18-CE20-0006 through the funding of the project MetaboLight (D.P.); the French National Research Agency through the funding of the Grenoble Alliance for Integrated Structural & Cell Biology GRAL project ANR-17-EURE-0003 (D.P., MAR-S., Y.Y., O.B., J.J., A.T., G.K., E.T., M.T., S.B., Y.C.); the French National Research Agency in the framework of the Investissements d'Avenir program ANR-15-IDEX-02, through the funding of the "Origin of Life" project of the Univ. Grenoble-Alpes (D.P., Y.Y., O.B.); the Prestige Marie-Curie co-financing grant PRESTIGE–2017-1-0028 (MAR-S.); the International Max Planck Research School "Primary Metabolism and Plant Growth" at the Max Planck Institute of Molecular Plant Physiology (M.A., Z.N.); the French National Research Agency through the funding of the project ProFI (Proteomics French Infrastructure, ANR-

10-INBS-08 (M.T., S.B., Y.C.); the U.S. National Science Foundation CAREER award MCB-1552522 (L.H.); the French National Research Agency (ANR-20-CE20-0027 for M.K. Y.L.-B. We would like to thank the following colleagues for their comments on the manuscript: John Christie, Roman Ulm, Arthur R. Grossman, Sebastian Westenhoff, Emanuel Sanz-Luque, and David Dauvillée. We also extend our thanks to Yalikunjiang Aizezi, Zhi-Yong Wang, and Maria Jimenez Palma for their assistance during the manuscript revision process.

## Author contributions

Conceptualization: Y.Y., D.P. Methodology: Y.Y., A.A.I., P.W.S., M.K., M.A., D.F., I.S., O.B. Investigation: Y.Y., A.A.I., M.K., A.T., M.A.R., G.K., E.T., M.T., S.B., Y.C., J.P.K., M.S., J.J., O.B. Visualization: Y.Y., A.A.I., P.W.S., M.A. Funding acquisition: D.P., L.M.H., P.H., Y.L.B., Z.N. Project administration: D.P. Supervision: D.P., L.M.H., P.H., Y.L.B., Z.N. Writing—original draft: Y.Y., D.P. Writing—review and editing: Y.Y., D.P. with contributions of all coauthors.

## Funding

## Competing interests

The authors declare no competing interests.
