## [Transparent Peer Review file · Nature Communications]

Phototropin connects blue light perception to starch metabolism in green algae

Corresponding Author: Professor Dimitris Petroustos

Version 0:

Reviewer comments:

Reviewer #1

(Remarks to the Author)

This is a fascinating manuscript in which the authors start uncovering how phototropin (a blue light photoreceptor) controls starch metabolism in *Chlamydomonas*. It is insightful because the role of phototropins in controlling metabolism is overall very poorly understood. There are well known indirect effects through which phototropins control carbon fixation by e.g. regulating the position of photosynthetic organs, the aperture of stomata and the position of chloroplasts. However, here the authors reveal a role of phototropin in light-regulated starch accumulation, which depends on a downstream protein kinase (PMSK1) and regulated mRNA levels of a starch metabolic enzyme (GAP1). How these different players (phototropin, PMSK1 and GAP1 mRNA levels) are connected at the molecular level remains to be discovered but the study clearly advances our understanding of light-regulated metabolism and identifies an important role for phototropin in the process. However, I have some issues with the model presented by the authors, clarifications are required. Moreover, the manuscript could be improved particularly in the discussion, because several directly relevant studies are either not discussed at all (e.g. the known control of starch metabolism by phototropins in *Arabidopsis* stomata) or poorly described (e.g. the mechanism by which *Arabidopsis* phot1 regulates NPH3 phosphorylation that precedes de-phosphorylation).

Comments

1. Page 6, lines 35-36. The authors report that *pmsk1* mutants accumulate lower levels of GAP1 mRNA in white, blue and red light and the effect of the mutant was strongest in red light. I have 2 questions here. 1) In the model presented on Fig. 5C PMSK1 negatively regulates GAP1 expression. So why would GAP1 mRNA levels be lower in the loss-of-function mutant? GAP1 levels should be higher if you eliminate a negative regulator. 2) It is difficult to understand why the difference between the mutant (*pmsk1*) and the WT is biggest in red light, a condition when phot1 is not active & PMSK1 not dephosphorylated (figure 3C). The authors have to provide some plausible explanation for their interesting data that is not explained by their model or in the text.
2. Page 6, lines 26, 27. The authors write that the phosphorylation state of PMSK1 S120 regulates starch metabolism through transcriptional regulation of GAP1. I have two problems with this statement. First the biggest difference between the WT and *pmsk1* mutant regarding GAP1 transcript abundance is in red light when PMSK1 phosphorylation is not regulated by phot1 (see my comment above). Second, the authors report GAP1 mRNA levels. A change in mRNA levels does not imply a change in GAP1 transcription, it could be due to a change in mRNA stability. Interestingly in *Arabidopsis* phot1 was shown to regulate mRNA stability (through unknown mechanisms), PMID: 12650626.
3. Page 3 lines 23-24. Not sure what exactly PHOT activity means in this phrase. I presume that this construct should be a constitutively active protein kinase but believe that the authors rather refer to its mutant complementation potential. Please clarify
4. Page 5, last phrase. The authors write that the mobility shift can be exclusively attributed to phosphorylation of residue 120. This is not correct. Indeed, the observed mobility shift requires phosphorylation of residue 120, but the observed change in mobility might be due to additional phosphorylation events that depend on the initial phosphorylation of S120
5. About the discussion. In *Arabidopsis* phot-dependent de-phosphorylation of NPH3 depends on phot-dependent phosphorylation of NPH3 (direct phosphorylation) that then triggers de-phosphorylation of NPH3. Do the authors envisage something similar here? this should be discussed taking into consideration recent work on the topic, e.g. PMID: 34675214 and PMID: 34675219. ref 30 does not show a link between the NPH3 phosphorylation state and phot1 ubiquitination. please rework that part of the discussion to clearly explain what is understood in *Arabidopsis*.
6. in reference 32 cited by the authors, there is rather compelling evidence for phot1 directly phosphorylating CBC1/2 in *Arabidopsis*. This seems like a rather important information given that PMSK1 is a related protein kinase. Also related to the

point raised above, do the authors envisage that in *Chlamydomonas* phot1 phosphorylates PMSK1 (similarly to the Arabidopsis CBC1/2 situation) at an unknown site and this then triggers subsequent PMSK1 dephosphorylation? This should be discussed.

7. In Arabidopsis, blue light in a phot dependent way triggers starch degradation in stomata, e.g. PMID: 26774787 and PMID: 32354788. It would be very relevant to compare and contrast phot-regulated starch metabolism in Arabidopsis and *Chlamydomonas* in the discussion.

Reviewer #2

(Remarks to the Author)

In this paper, Yuan et al. explored the relationship between light quality and starch metabolism in the model green alga *Chlamydomonas reinhardtii*. It has long been observed that *C. reinhardtii* exhibits lower starch content under blue light conditions and higher starch content under red light conditions. However, the precise mechanism underlying this phenomenon has remained largely elusive. Yuan et al. presented, for the first time, a connection between the blue light photoreceptor Phototropin (PHOT) and starch metabolism in *Chlamydomonas*. They identified downstream phosphorylation events and associated factors. Using phosphoproteomics, they discovered a novel protein, PMSK1, which is dephosphorylated by PHOT in a blue light-dependent manner. Dephosphorylated PMSK1 subsequently inhibits GAP1, resulting in reduced starch accumulation. The authors suggest that blue light inhibits starch accumulation, contrary to red light inducing it in green algae. While PHOT's role in regulating stomatal opening via guard cell starch degradation has been documented in the higher plant Arabidopsis, its involvement in starch metabolism in *Chlamydomonas* represents a novel finding with a mechanism distinct from that in Arabidopsis.

The paper comprises meticulously designed and challenging experiments on green algae, lending high reliability and robustness to their conclusions. It is lucidly written and easily comprehensible. Nonetheless, I would appreciate further elaboration or clarification on certain experiments.

- 1) phot is high light sensitive due to its compromised LHCSR3 expression. Starch has been reported could play a protective role under high light conditions. Is the high starch content to some degree linked to the deficiency in photoprotection? The simplest and easiest way to test this would be to reintroduce LHCSR3 into the PHOT mutant and check the starch content.
- 2) GAP1 (GAPDH) is one of the most important enzymes in the Calvin-Benson cycle (CBC). Interestingly to note that GAPDH alone affects starch accumulation greatly in *Chlamydomonas*. I am wondering if the authors have performed metabolomics to check the CBC and starch synthesis intermediates. If yes, have any differences between WT and phot mutants been found?
- 3) How did the authors pick up PMSK1 based on the phosphoproteomic data? There should be multiple ones fitting the patterns.
- 4) PMSK1 resides in the flagella, plasma membrane, and cytosol, while GAP1 is localized in the chloroplast. Does the phosphorylation of PMSK1 lead to a change in its location, which might regulate GAP1? As phosphorylation-induced localization changes are very common in higher plants. Besides, how does PMSK1 inhibit GAP1?
- 5) The authors mentioned that PMSK1 is homologous to Arabidopsis CBC1/HT1. In Arabidopsis, PHOT1 interacts with and phosphorylates CBC1. Moreover, as the authors wrote, NPH3 is dephosphorylated in a PHOT-dependent manner by an unknown phosphatase and then interacts with PHOT1 to trigger the ubiquitination of PHOT1. PMSK1 appears to be a similar case to NPH3. Did the authors check if PMSK1 interacts with PHOT? Perhaps PMSK1 is not a substrate but an interactor. If so, does mutating S120 affect the interaction? Although this experiment is not necessary for this manuscript and will not affect the manuscript results, it would be beneficial for further explanation and research.

Some minor points:

- 1) How do the authors know that D442 of PMSK1 is the ATP-binding site? Is this information derived from homology proteins or AlphaFold prediction? It would be helpful to include a brief explanation.
- 2) It would be great if the authors could provide more information about the generation of the pmsk1 and phot pmsk1 mutants, such as the insertion site sequence information or PCR or qPCR data.
- 3) There are typos and grammar errors. Should be gone through for corrections.

Reviewer #3

(Remarks to the Author)

The authors described an important finding that links blue light perception with starch accumulation in *Chlamydomonas*. They found blue light suppresses starch accumulation and the suppression is mediated mainly by the blue light receptor PHOTOTROPIN (PHOT). Through functional proteomics and phosphoproteomics approach, they found that PHOT suppress expression of GAP1, which is highly important for starch accumulation, through dephosphorylation of S120 site on the Ser/Thr protein kinase PMSK1, although the phosphatase responsible for the dephosphorylation was not identified. Overall, the manuscript was well written and the findings are highly novel, the data is solid and the conclusions are well supported by the data. Nevertheless, minor issues are still needed to be addressed.

1. In continuous light, the relative abundance of GAP1 in phot mutant is about 2 orders of magnitude higher than that in the WT (Fig. 2B). However, in Fig. 2C-D, mRNA abundance of GAP1 reached the highest (and nearly the same, 102) level in both WT and the phot mutant at the time point 12 h, so if the cells are continued to be incubated in light (continuous light), what will happen about the GAP1 mRNA level?
2. Lines 36-37: "GAP1 mRNA accumulation profile was identical in white and blue light, but reached maximal levels at the middle of the light phase under red light conditions (Fig. 2c)" The statement is not accurate, because GAP1 mRNA level did not reach the maximal level at the middle of the light phase.
3. In Fig.1C, the percentage of cell population showing the TEM pattern need to be reported.
4. In Fig. 2A, in addition to GAP1, other differentially abundant proteins in the phot mutant such as those involved in photosynthesis and light harvesting may also contribute to starch accumulation. It seems that these proteins were simply ignored in the study. The authors need as least to discuss this.
5. In supplemental Fig 12, cluster E, how many phosphosites were included in this cluster, what was the rational to choose the S120 from PMSK1 for the follow-up work?
6. The number of identified phosphopeptides (1119) is rather too small, considering that more than 10,000 phosphosites can be identified from *Chlamydomonas* using single-shot proteomics (Duan X, Mol Plant, 2024). The authors used TripleTOF 5600 for the identification of phosphoproteome but more advanced QE HF for the study of global proteome, is there any reason for this because it is better to use QE HF for phosphoproteomics.
7. In supplementary Fig. 14b, there was a mislabeling, wt should be changed to phot.
8. In supplemental Fig 16, the percentage of cell population showing the TEM pattern need to be reported.
9. In supplementary Fig. 18, the authors aimed to demonstrate that the observed regulatory role of PMSK1 on starch metabolism is independent of its kinase activity. However, in page 6, line 31-32 of the manuscript, the authors stated that PMSK1 fulfills the observed regulatory role on starch metabolism through its kinase activity. Obviously the data contradicts with the statement.
10. Page 7, line 25, Fig. 1C should be Fig. 1D.

Version 1:

Reviewer comments:

Reviewer #2

(Remarks to the Author)

The authors have addressed the comments in a reasonable way. I have no additional comments to this revised revision

Reviewer #3

(Remarks to the Author)

The authors fully addressed my concerns and made appropriate clarifications and revisions. The reviewer believes that the quality of the research and manuscript per se is deserved to be published in NC.

Reviewer #4

(Remarks to the Author)

The manuscript by Yuan et al. describes the involvement of blue-light-induced phototropin in starch metabolism in *Chlamydomonas*. The finding is quite new and interesting. However, the data and the description are insufficient to publish this paper in this journal. Authors should add some data, consider previous findings in other algal and plant species, and thoroughly revise the manuscript, in particular Introduction and Discussion sections.

Concerns

Authors analyzed only the regulation of GAP1 mRNA to reveal the molecular mechanism underlying phot-mediated regulation of starch accumulation, the effect of GAP1 overexpression and downregulation on starch accumulation was quite mild compared to phot and phot-kin strains. Do authors think that phot strain is defective only in starch synthesis but is normal in starch breakdown? In another Chlorophyte *Chlorella*, the blue-light-induced starch breakdown is well-known (Kowallik and Gaffron, *Planta*, Vol. 69: 92-95, 1966; Kamiya and Miyachi, *Plant Cell Physiol.*, Vol. 15: 927-937, 1974; Kowallik, *Annu Rev Plant Physiol*, Vol. 33: 51-72, 1982). It was hypothesized that this response is regulated by the 120-kDa blue-light photoreceptor localized on the plasma membrane (Matschke et al., *Photochem Photobiol*, Vol. 66: 128-132, 1997; Kamiya and Saitoh, *Physiol. Plant*, Vol. 116: 248-254, 2002). Authors compared their results with only those in *Arabidopsis* stomata, they must not ignore the results in *Chlorella* if their research focus on photosynthetic "microalgae".

"PHOTOTROPIN (PHOT)" means a non-functional phototropin apoprotein and thus is usually not used. When the functions of phototropins are described, authors should use "phototropin (phot)" that means phototropin holoprotein or "italicized PHOTOTROPIN (PHOT)" that means phototropin gene. Authors should use correct nomenclature of phototropins (Briggs et al., *Plant Cell*, Vol. 13: 993-996, 2001).

There is no introduction of *Chlamydomonas* photoreceptors in Introduction section.

Authors should indicate what kind of statistical tests was performed in the legends of each Figure.

Supplementary Figure 1

It is still possible that decrease in starch levels could result from decrease in light intensity of red light from 50 to 45 μE . Furthermore, it is not clear whether the suppression of beneficial effect of red light was specific to blue light and which wavelength of light is effective in starch accumulation under white light condition. Authors should analyze starch levels in cells exposed to only blue light and other wavelength light (green and far-red). Because authors analyzed only blue-light effect, the involvement of channelrhodopsins in starch accumulation remained to be determined.

Page 3, lines 25-29

Authors should mention not only very thick starch sheaths around pyrenoid but also the ectopic starch accumulation apart from pyrenoid in phot mutant, because authors put two arrows on a photograph for phot mutant in Fig. 1C. Because there is no introduction of starch accumulation pattern in *Chlamydomonas* in Introduction section, most readers would not understand it. At least, I do not know what is the difference between WT and phot in Supplementary Fig. 4. There is no data for phot-C1 strain in Supplementary Fig. 4 although that was mentioned in the text (line 28).

Page 5, lines 25-36

I have no idea why authors focused on cluster E. "To identify missing components in the phototropin-mediated signalling pathway", we will usually screen the direct substrate of phototropins with phosphoproteomics and thus focus on clusters M and O. At least, the rationale that authors focused on cluster E should be described here.

Page 6, lines 3-5

In Fig. 3c and d, the unphosphorylated band can be observed 20 min after blue light irradiation in phot strain. The band is very weak, but significant compared to red-light irradiated samples. Authors should revise the sentence as follow; whereas in the phot background, dephosphorylation was hardly observed under blue light (Fig. 3c). No dephosphorylation occurred under red light in both strains (Fig. 3c).

The data in Supplementary Fig. 16 is very important and thus should be transferred to Fig. 3.

Page 6, lines 12-38

Authors should refer to Supplementary Fig. 4 here.

Page 7, lines 10-34

This part should be transferred to Discussion section.

Page 8, lines 1-2

Our work establishes PMSK1 as the first kinase described in to mediate phot-dependent signalling in "*Chlamydomonas* (or green alga)"

Page 8, lines 13 -14

References 30 and 31 are not papers that indicate that "NPH3 may function as a substrate adaptor in a CULLIN3-base E3 ubiquitin ligase complex, targeting phot1 for ubiquitination. Moreover, the role of NPH3 in phot1 ubiquitination is controversial in the field of phototropin research. Thus, the sentence should be deleted.

Page 8, lines 14 -15

This sentence leads to misunderstanding. Completely dephosphorylated NPH3 is inactive and partially phosphorylated NPH3 is active (Christie et al., *Plant Physiol*, Vol. 176: 1015-1024, 2018).

Page 8, lines 21-25

Reference 33 is a paper on CBC1/2, but not for HT1. Authors should cite an appropriate paper on HT1.

Version 2:

Reviewer comments:

Reviewer #4

(Remarks to the Author)

The authors sincerely addressed all of my concerns and thus I have no additional concerns in the revised manuscript.

AUTHOR REBUTTALS TO REVIEWERS' COMMENTS:

We thank the reviewers for their time in reviewing this manuscript and their positive comments on the novelty and importance of this work. We also thank the reviewers for the constructive comments and suggestions that have helped us improve this manuscript.

A point-by-point response to the reviewers' comments is presented below, in blue.

REVIEWER COMMENTS

Reviewer #1 (Remarks to the Author):

This is a fascinating manuscript in which the authors start uncovering how phototropin (a blue light photoreceptor) controls starch metabolism in *Chlamydomonas*. It is insightful because the role of phototropins in controlling metabolism is overall very poorly understood. There are well known indirect effects through which phototropins control carbon fixation by e.g. regulating the position of photosynthetic organs, the aperture of stomata and the position of chloroplasts. However, here the authors reveal a role of phototropin in light-regulated starch accumulation, which depends on a downstream protein kinase (PMSK1) and regulated mRNA levels of a starch metabolic enzyme (GAP1). How these different players (phototropin, PMSK1 and GAP1 mRNA levels) are connected at the molecular level remains to be discovered but the study clearly advances our understanding of light-regulated metabolism and identifies an important role for phototropin in the process. However, I have some issues with the model presented by the authors, clarifications are required. Moreover, the manuscript could be improved particularly in the discussion, because several directly relevant studies are either not discussed at all (e.g. the known control of starch metabolism by phototropins in *Arabidopsis* stomata) or poorly described (e.g. the mechanism by which *Arabidopsis* phot1 regulates NPH3 phosphorylation that precedes de-phosphorylation).

We sincerely thank the reviewer for the positive feedback on our manuscript and for sharing our enthusiasm for these results. We have revised our proposed model, improved the discussion section of the manuscript and addressed the specific comments as follows.

Comments

1. Page 6, lines 35-36. The authors report that *pmsk1* mutants accumulate lower levels of GAP1 mRNA in white, blue and red light and the effect of the mutant was strongest in red light. I have 2 questions here. **1)** In the model presented on Fig. 5C PMSK1 negatively regulates GAP1 expression. So why would GAP1 mRNA levels be lower in the loss-of-function mutant? GAP1 levels should be higher if you eliminate a negative regulator. **2)** It is difficult to understand why the difference between the mutant (*pmsk1*) and the WT is biggest in red light, a condition when phot1 is not active & PMSK1 not dephosphorylated (figure 3C). The authors have to provide some plausible explanation for their interesting data that is not explained by their model or in the text.

1.1. Thank you for your insightful comment, which touches on a point we have been carefully considering for some time. Based on our data, we initially proposed in our model (**Fig. 6**) that the non-phosphorylated form of PMSK1 at S120 acts as an inhibitor of GAP1 mRNA accumulation. However, in light of your comment, we have revised this conclusion.

We have now indicated that PMSK1 acts as an activator of *GAP1* mRNA accumulation, and its activity is regulated by the phosphorylation status of S120 in response to light quality. When PMSK1 is phosphorylated (as in the S120D strain), it becomes active and promotes *GAP1*

mRNA accumulation. When it is not phosphorylated (as in the S120A strain), PMSK1 is inactive, and GAP1 induction does not occur—though it may still be influenced by other signals such as circadian or diurnal rhythms. **In red light:** The dominant form of PMSK1 is the phosphorylated one (PMSK1-P; **Fig. 3c**), driving high *GAP1* mRNA accumulation (**Fig. 4a**). **In white and blue light:** Dephosphorylation of active PMSK1-P results in an accumulation of unphosphorylated PMSK1 (PMSK1-U; **Fig. 3c**), reducing the relative abundance of PMSK1-P within the total PMSK1 pool. This shift toward PMSK1-U decreases the proportion of PMSK1-P, leading to reduced *GAP1* mRNA levels (**Fig. 4a**). **In the *pmsk1* mutant,** *GAP1* mRNA levels (**Fig. 5a**) are similar to those in WT/PMSK1-S120A (unphosphorylated form; **Fig. 4a**), showing lower expression at 12 hours. However, there is still regulation in red light, which is abolished in the *phot pmsk1* double mutant (**Fig. 5a**), suggesting the existence of other repressors of *GAP1* that are controlled by blue light independently of PMSK1, although PMSK1 appears to play a major role.

Our model is now presented as a standalone Fig. 6. The model has been updated to summarize all findings and hypotheses presented in our manuscript, namely:

- 1. The activating role of PMSK1-P on GAP1.** Supporting evidence: Figs. 3, 4 and 5; as detailed above.
- 2. A PHOT-dependent but PMSK1-independent repression of GAP1.** Supporting evidence: *GAP1* in *pmsk1* is still regulated by red light and this regulation is lost in the *phot pmsk1* double mutant (**Fig. 5a**).
- 3. “Blue light-responsive protein(s)” inhibiting starch accumulation independently of both PHOT and GAP1.** Supporting evidence: *phot* cells accumulate more starch under red light (**Fig. 1d**), despite *GAP1* mRNA levels being similarly high across all light conditions in *phot* cells (**Fig. 2d**).
- 4. “Blue light-responsive protein(s)” inhibiting GAP1 mRNA levels in a PHOT-independent manner.** Supporting evidence: in the *phot-kin* strain in which PHOT is devoid of the photosensory LOV domains and is always active regardless of light quality, accumulated very low levels of *GAP1* mRNA (**Fig. 2g**). However, in *phot-kin* cells grown under red light, *GAP1* mRNA levels are higher at 6 hours compared to cells grown in white or blue light (**Fig. 2g**). Despite the increased *GAP1* mRNA in red light this was not sufficient for overaccumulation of starch (**Fig. 1d**). This indicates that additional regulators, beyond PHOT, may be involved in repressing *GAP1* in a blue light-dependent manner.

1.2. Based on the reasoning outlined above, under red light, the active PMSK1-P is dominant. In contrast, under white or blue light, where PHOT is active promoting the de-phosphorylation of PMSK1, there is less PMSK1-P in mixture with PMSK1-U. As a result, the mutation in PMSK1 has the greatest effect on *GAP1* mRNA levels in red light, where PMSK1-P is most prevalent. To better represent the data, we replaced **Fig. 5a**, which showed *GAP1* levels only at t=6h, with Supplementary Fig. 19, which displays the full diurnal rhythmicity of *GAP1* mRNA across the different strains.

2. Page 6, lines 26, 27. The authors write that the phosphorylation state of PMSK1 S120 regulates starch metabolism through transcriptional regulation of *GAP1*. I have two problems with this statement. First the biggest difference between the WT and *pmsk1* mutant regarding *GAP1* transcript abundance is in red light when PMSK1 phosphorylation is not regulated by *phot1* (see my comment above). Second, the authors report *GAP1* mRNA levels. A change in mRNA levels does not imply a change in *GAP1* transcription, it could be due to a change in mRNA

stability. Interestingly in *Arabidopsis phot1* was shown to regulate mRNA stability (through unknown mechanisms), PMID: 12650626.

We have addressed the reviewer's first point above (see point 1.2). Regarding the second point, we agree that measuring *GAP1* mRNA levels does not allow us to directly conclude changes in *GAP1* transcription, as variations in mRNA levels can also result from changes in mRNA stability rather than transcriptional activity (as it has been reported to be the case in *Arabidopsis*). In light of this, we have rephrased our statement to: "the phosphorylation state of S120 of PMSK1 controls starch metabolism through the regulation of *GAP1* mRNA levels" to accurately reflect our findings without assuming transcriptional changes.

In a similar context, our previous work on PHOT regulation of *LHCSR3* mRNA accumulation demonstrated that PHOT-dependent signaling influenced *LHCSR3* transcript levels, primarily through the activation of transcription and partially via transcript stabilization (<https://www.nature.com/articles/nature19358>). This supports the idea that PHOT can influence both transcriptional activity and mRNA stability, contributing to changes in mRNA levels and underscoring the complexity of the underlying regulatory mechanisms.

3. Page 3 lines 23-24. Not sure what exactly PHOT activity means in this phrase. I presume that this construct should be a constitutively active protein kinase but believe that the authors rather refer to its mutant complementation potential. Please clarify.

Thank you for pointing this out. Yes, when only the kinase domain of PHOT is introduced back into the *phot* mutant, *phot-kin* will behave as if PHOT is always activated, independently on the light-spectrum. We have modified the text accordingly.

4. Page 5, last phrase. The authors write that the mobility shift can be exclusively attributed to phosphorylation of residue 120. This is not correct. Indeed, the observed mobility shift requires phosphorylation of residue 120, but the observed change in mobility might be due to additional phosphorylation events that depend on the initial phosphorylation of S120

This is a very good point, thank you. We have revised the text accordingly (p. 6, lines 7-11) as shown below:

"Interestingly, the observed blue light dependent mobility shift of PMSK1-FLAG (**Fig. 3c**) was abolished when S120 was substituted by an alanine (A) or an aspartic acid (D) residue (**Supplementary Fig. 16**). These data indicate that phosphorylation at residue S120 is essential for the shift. However, this shift may not be solely attributed to S120 phosphorylation, but rather to additional phosphorylation events that depend on the initial phosphorylation of S120."

5. About the discussion. In *Arabidopsis phot*-dependent de-phosphorylation of NPH3 depends on phot-dependent phosphorylation of NPH3 (direct phosphorylation) that then triggers de-phosphorylation of NPH3. Do the authors envisage something similar here? this should be discussed taking into consideration recent work on the topic, e.g. PMID: 34675214 and PMID: 34675219. ref 30 does not show a link between the NPH3 phosphorylation state and *phot1* ubiquitination. please rework that part of the discussion to clearly explain what is understood in *Arabidopsis*.

We have addressed comment 5 together with comment 6, please see below.

6. in reference 32 cited by the authors, there is rather compelling evidence for phot1 directly phosphorylating CBC1/2 in Arabidopsis. This seems like a rather important information given that PMSK1 is a related protein kinase. Also related to the point raised above, do the authors envisage that in *Chlamy* phot1 phosphorylates PMSK1 (similarly to the Arabidopsis CBC1/2 situation) at an unknown site and this then triggers subsequent PMSK1 dephosphorylation? This should be discussed.

We have modified the discussion section according to reviewer's comments 5 and 6, referring to the PMID: 34675214 and PMID: 34675219. Please see the updated NPH3-related text in **page 8, lines 6-18**, followed by lines 19-28 addressing the PMSK1-CBC1/2-related comment in **lines 19-28**.

Lines 6-18: How PHOT, a kinase, is involved in the dephosphorylation of PMSK1 is an intriguing question that needs further investigation. One of the possibilities is that a protein phosphatase (indicated PPase in **Fig. 6**) is a missing component in our proposed model; PHOT may mediate the activity of this PPase to dephosphorylate S120 of PMSK1. In *Arabidopsis*, which encodes two phototropins, PHOT1 and PHOT2, blue light induces the phosphorylation of NPH3 (NON-PHOTOTROPIC HYPOCOTYL3) at serine 744 (S744) in a PHOT1-dependent manner. This phosphorylation allows NPH3 to associate with 14-3-3 proteins, followed by its dephosphorylation. NPH3 may function as a substrate adaptor in a CULLIN3-based E3 ubiquitin ligase complex, targeting PHOT1 for ubiquitination^{30,31}. The dephosphorylation of NPH3 is associated with enhanced phototropic responsiveness in de-etiolated seedlings. Whether *Chlamydomonas* PHOT operates via a similar mechanism—interacting with and phosphorylating PMSK1 prior to its dephosphorylation at S120—remains to be determined.

Lines 19-28: PMSK1 is found to belong to a family of Serine/threonine-protein kinases (named PMSK-like family, see material and methods) which is conserved in green algae and vascular plants. When searching for PMSK1-like sequences in the *Arabidopsis* genome in the NCBI database³², HIGH LEAF TEMPERATURE 1 (HT1) and CONVERGENCE OF BLUE LIGHT AND CO₂ 1/2 (CBC1/2) are found in addition to the PMSK-like Arabidopsis members (**Supplementary Figs. 22-25, Supplementary Text**), proteins that have been shown to mediate responses to CO₂ and blue light in *Arabidopsis*³³. Yet, while *Arabidopsis* CBC1/2/HT1 act to stimulate stomatal opening by inhibiting S-type anion channels, our findings demonstrate that the regulatory function of its *Chlamydomonas* counterpart PMSK1 controls starch metabolism through the transcriptional regulation of *GAP1*.

7. In Arabidopsis, blue light in a phot dependent way triggers starch degradation in stomata, e.g. PMID: 26774787 and PMID: 32354788. It would be very relevant to compare and contrast phot-regulated starch metabolism in Arabidopsis and *Chlamy* in the discussion.

We have added this to the discussion section.

Reviewer #2 (Remarks to the Author)

In this paper, Yuan et al. explored the relationship between light quality and starch metabolism in the model green alga *Chlamydomonas reinhardtii*. It has long been observed that *C. reinhardtii* exhibits lower starch content under blue light conditions and higher starch content under red light conditions. However, the precise mechanism underlying this phenomenon has remained largely elusive. Yuan et al. presented, for the first time, a connection between the blue light photoreceptor Phototropin (PHOT) and starch metabolism in *Chlamydomonas*. They identified downstream phosphorylation events and associated factors. Using phosphoproteomics, they discovered a novel protein, PMSK1, which is dephosphorylated by PHOT in a blue light-dependent manner. Dephosphorylated PMSK1 subsequently inhibits GAP1, resulting in reduced starch accumulation. The authors suggest that blue light inhibits starch accumulation, contrary to red light inducing it in green algae. While PHOT's role in regulating stomatal opening via guard cell starch degradation has been documented in the higher plant *Arabidopsis*, its involvement in starch metabolism in *Chlamydomonas* represents a novel finding with a mechanism distinct from that in *Arabidopsis*.

The paper comprises meticulously designed and challenging experiments on green algae, lending high reliability and robustness to their conclusions. It is lucidly written and easily comprehensible. Nonetheless, I would appreciate further elaboration or clarification on certain experiments.

We appreciate the reviewer's positive comments on our manuscript.

1. phot is high light sensitive due to its compromised LHCSR3 expression. Starch has been reported could play a protective role under high light conditions. Is the high starch content to some degree linked to the deficiency in photoprotection? The simplest and easiest way to test this would be to reintroduce LHCSR3 into the PHOT mutant and check the starch content.

The reviewer raises an important point. We have tested this by using the *pcsr3* mutant generated in a previous study of ours where we described the link between PHOT and LHCSR3 (*Nature* volume 537, pages563–566, 2016) The *pcsr3* strain is a *phot* mutant line expressing an additional copy of the *LHCSR3.1* gene under the control of a constitutive *PsaD* promoter. The *pcsr3* strain showed a similar starch content to the *phot* mutant under the experimental conditions we used throughout the manuscript (**Rebuttal Fig.1**). However, we have opted not to include this information in the final manuscript for two primary reasons: First, the conditions examined in our study involve low light, which does not induce LHCSR3 expression to high levels. Second, as the focus of this manuscript is specifically on the role of PHOT in starch metabolism, we felt that including this detail would shift the focus away from the main findings. We hope the reviewer understands our rationale for this decision.

Rebuttal Fig.1. Starch content of WT (*cw15-302*), *phot* and *pcsr3* grown under continuous white light at 50µE. Data are represented as mean ± SD (n = 3 biologically independent samples). Asterisks indicated the p-values (****, p < 0.0001; ns, not significant).

2. GAP1 (GAPDH) is one of the most important enzymes in the Calvin-Benson cycle (CBC). Interestingly to note that GAPDH alone affects starch accumulation greatly in *Chlamydomonas*. I am wondering if the authors have performed metabolomics to check the CBC and starch synthesis intermediates. If yes, have any differences between WT and *phot* mutants been found?

Thank you for this excellent suggestion. In fact, we have recently collected samples for both metabolomics and transcriptomics throughout the diurnal cycle in WT and *phot* mutants, with the goal of gaining deeper insights into PHOT-dependent signaling and its integration with diurnal and circadian regulation. The analysis of these experiments is ongoing, and we plan to present these findings in a future publication.

3. How did the authors pick up PMSK1 based on the phosphoproteomic data? There should be multiple ones fitting the patterns.

Our phosphoproteomics data identified several promising candidates, with PMSK1 standing out due to its similarity to the known *Arabidopsis* PHOT substrates, CBC1 and HT1. As noted in our manuscript, S120 of PMSK1 exhibits PHOT-dependent dephosphorylation, which led us to select PMSK1 for further investigation. In the manuscript, we first discuss the functional role of PMSK1 before addressing its evolutionary connection to CBC1 and HT1.

4) PMSK1 resides in the flagella, plasma membrane, and cytosol, while GAP1 is localized in the chloroplast. Does the phosphorylation of PMSK1 lead to a change in its location, which might regulate GAP1? As phosphorylation-induced localization changes are very common in higher plants. Besides, how does PMSK1 inhibit GAP1?

Thank you for asking this excellent question. Our data suggest that the effect of PMSK1 on GAP1 occurs at the transcriptional level, suggesting that PMSK1-derived signalling likely reaches the nucleus to regulate *GAP1* mRNA accumulation. Exactly how this signalling process unfolds remains an open question that we plan to investigate further. In this context, direct regulation of the GAP1 protein or enzyme by PMSK1 seems less likely.

We agree with the reviewer that phosphorylation-induced localization changes are indeed common in plants. In our experiments, we observed similar shifts when expressing different phosphomimetic variants of PMSK1 in the wild-type strain (see **Rebuttal Fig. 2**). Specifically, PMSK1^{S120D} is localized in the cytosol, whereas PMSK1^{S120A} interestingly localizes to the chloroplast. However, the biological significance of this localization shift remains to be determined.

A potential question is whether the chloroplast localization of PMSK1^{S120A} (or, conversely, its absence from the cytosol) could explain the low GAP1 expression observed in this strain. We can rule this out because, in the *phot* mutant, although PMSK1^{S120D} shows a minor chloroplast signal, yet both PMSK1^{S120D} and PMSK1^{S120A} primarily localize in the cytosol. Despite this, they have opposing effects on GAP1 mRNA levels: PMSK1^{S120D} is associated with high GAP1 expression, whereas PMSK1^{S120A} results in low GAP1 expression.

Rebuttal Fig.2. PMSK1^{S120D} and PMSK1^{S120A} subcellular localization. Representative confocal fluorescent microscopy images of PMSK1^{S120D}-FLAG-Venus or PMSK1^{S120A}-FLAG-Venus expressed in WT or *phot*. In green; signal from Venus fluorescent protein, in purple; chlorophyll autofluorescence.

5) The authors mentioned that PMSK1 is homologous to Arabidopsis CBC1/HT1. In Arabidopsis, PHOT1 interacts with and phosphorylates CBC1. Moreover, as the authors wrote, NPH3 is dephosphorylated in a PHOT-dependent manner by an unknown phosphatase and then interacts with PHOT1 to trigger the ubiquitination of PHOT1. PMSK1 appears to be a similar case to NPH3. Did the authors check if PMSK1 interacts with PHOT? Perhaps PMSK1 is not a substrate but an interactor. If so, does mutating S120 affect the interaction? Although this experiment is not necessary for this manuscript and will not affect the manuscript results, it would be beneficial for further explanation and research.

We thank the reviewer for pointing out this important aspect and agree that examining the interaction between PHOT and PMSK1 would benefit further research.

Some minor points:

1) How do the authors know that D442 of PMSK1 is the ATP-binding site? Is this information derived from homology proteins or AlphaFold prediction? It would be helpful to include a brief explanation.

D is part of the DFG/DFD motif that is conserved for ATP binding in kinases, as described in Ung and Schlessinger 2015 (doi: 10.1021/cb500696t).

2) It would be great if the authors could provide more information about the generation of the *pmsk1* and *phot pmsk1* mutants, such as the insertion site sequence information or PCR or qPCR data.

Such information is now provided in **Supplementary Figure. 20**.

3) There are typos and grammar errors. Should be gone through for corrections.

We have checked the entire manuscript and corrected all typos and errors.

Reviewer #3 (Remarks to the Author):

The authors described an important finding that links blue light perception with starch accumulation in *Chlamydomonas*. They found blue light suppresses starch accumulation and the suppression is mediated mainly by the blue light receptor PHOTOTROPIN (PHOT). Through functional proteomics and phosphoproteomics approach, they found that PHOT suppress expression of GAP1, which is highly important for starch accumulation, through dephosphorylation of S120 site on the Ser/Thr protein kinase PMSK1, although the phosphatase responsible for the dephosphorylation was not identified. Overall, the manuscript was well written and the findings are highly novel, the data is solid and the conclusions are well supported by the data. Nevertheless, minor issues are still needed to be addressed.

We thank the reviewer for the positive comments. Please see our detailed response below:

1. In continuous light, the relative abundance of GAP1 in *phot* mutant is about 2 orders of magnitude higher than that in the WT (Fig. 2B). However, in Fig. 2C-D, mRNA abundance of GAP1 reached the highest (and nearly the same, 102) level in both WT and the *phot* mutant at the time point 12 h, so if the cells are continued to be incubated in light (continuous light), what will happen about the GAP1 mRNA level?

Thank you for this insightful question. As shown in Figure 2B, under continuous light conditions, *GAP1* mRNA levels in the *phot* mutant are significantly higher compared to the WT, indicating that PHOT plays a key regulatory role in suppressing *GAP1* expression in such conditions. In contrast, under the 12-hour light/dark cycle (Figures 2C-D), both WT and the *phot* mutant exhibit similar peak levels of GAP1 mRNA at the 12-hour time point, suggesting that *GAP1* expression in both strains is regulated by diurnal rhythms.

These observations imply that under diurnal light conditions, *GAP1* mRNA is influenced not only by PHOT activity but also by circadian or diurnal factors. If the cells were incubated in continuous light beyond the 12-hour time point, based on our current understanding and the data from Figure 2B, we would expect GAP1 mRNA levels in the *phot* mutant to remain elevated, continuing to exceed those of the WT. Without functional PHOT, the suppression of GAP1 mRNA does not occur, allowing the levels to stay high.

In contrast, for the WT, we anticipate that *GAP1* mRNA levels would gradually decrease after the peak at 12 hours. PHOT, which remains active under continuous light, would continue to suppress *GAP1* expression, leading to a divergence in mRNA levels between the WT and the *phot* mutant. The *phot* mutant would maintain high levels of GAP1 mRNA, while levels in the WT would drop.

It is important to note that the data in Figure 2B represent cells where the circadian/diurnal rhythm has been disrupted due to their continuous exposure to light throughout their culture history. This distinction helps explain the difference in GAP1 regulation between continuous light and diurnal light conditions.

2. Lines 36-37: “GAP1 mRNA accumulation profile was identical in white and blue light, but reached maximal levels at the middle of the light phase under red light conditions (Fig. 2c)” The statement is not accurate, because GAP1 mRNA level did not reach the maximal level at the middle of the light phase.

Thank you for pointing this out. We revised the text to describe the graph more accurately.

3. In Fig.1C, the percentage of cell population showing the TEM pattern need to be reported.

8. In supplemental Fig 16, the percentage of cell population showing the TEM pattern need to be reported.

In our TEM results, almost all the cells we observed showed a similar phenotype. We included the TEM images of groups of cells from different strains in a new **Supplementary Fig. 4**.

4. In Fig. 2A, in addition to GAP1, other differentially abundant proteins in the phot mutant such as those involved in photosynthesis and light harvesting may also contribute to starch accumulation. It seems that these proteins were simply ignored in the study. The authors need at least to discuss this.

Thank you for your valuable suggestion. Indeed, our comparative proteomics analysis revealed a significant upregulation of proteins involved in photosynthesis and light harvesting in the *phot* mutant (e.g., LHCSR1, LHCA3, LHCA7, LHCA9, LHCBM5, PTOX2). However, GAP1, being the third most upregulated protein in *phot* and given its key metabolic role, naturally stood out and became a focal point of our investigation, proving to be a well-informed choice.

5. In supplemental Fig 12, cluster E, how many phosphosites were included in this cluster, what was the rationale to choose the S120 from PMSK1 for the follow-up work?

The number of phosphopeptides in each cluster is indicated in parentheses in Supplementary Fig. 13. Cluster E contains a total of 84 phosphopeptides. Based on our phosphoproteomics data, we identified several intriguing candidates. Among them, PMSK1 stood out due to its homology to CBC1 and HT1, known substrates of PHOT in Arabidopsis. As described in our manuscript, S120 of PMSK1 exhibited PHOT-dependent dephosphorylation, which led us to select PMSK1 for further investigation.

6. The number of identified phosphopeptides (1119) is rather too small, considering that more than 10,000 phosphosites can be identified from Chlamydomonas using single-shot proteomics (Duan X, Mol Plant, 2024). The authors used TripleTOF 5600 for the identification of phosphoproteome but more advanced QE HF for the study of global proteome, is there any reason for this because it is better to use QE HF for phosphoproteomics.

We appreciate the reviewer's observation regarding the number of identified phosphopeptides. To clarify, the phosphoproteome and global proteome experiments were conducted at different times and in separate laboratories. Consequently, the equipment used for each experiment was based on the resources available in those facilities, which may have contributed to variations in depth of coverage. Despite these differences, we are confident in the quality of the data generated. While the number of phosphopeptides identified in our study (1119) is smaller compared to some other studies, the results are robust and offer valuable insights.

7. In supplementary Fig. 14b, there was a mislabeling, wt should be changed to phot.

We thank the reviewer for the careful reading. In **Supplementary Fig. 15b**, "wt" indicates that the PMSK1 in *phot*/PMSK1-FLAG is the WT background (non-mutated version).

9. In supplementary Fig. 18, the authors aimed to demonstrate that the observed regulatory role of PMSK1 on starch metabolism is independent of its kinase activity. However, in page 6, line 31-32 of the manuscript, the authors stated that PMSK1 fulfills the observed regulatory role on

starch metabolism through its kinase activity. Obviously, the data contradicts with the statement.

Thank you for highlighting this point. Our intention was indeed to demonstrate that the regulatory role of PMSK1 on starch metabolism *is* dependent on its kinase activity. We apologize if this was unclear. In the manuscript, we state:

“To investigate whether this role is mediated by the kinase activity of PMSK1 we combined an inactive-kinase version of PMSK1, in which Asp-442 was replaced with Asn (D442N) to inactivate the ATP-binding site, with the phosphomimetic mutation S120D or the non-phosphorylatable mutation S120A. Neither case resulted in an effect on starch content (Supplementary Fig. 19). Thus, PMSK1 fulfills the observed regulatory role on starch metabolism through its kinase activity.”

Our data support this conclusion, as neither S120-related mutation impacted starch content in the kinase-inactive version of PMSK1. We hope this clarification addresses the perceived contradiction.

We slightly modified the text in the manuscript to ensure our message is clear:

“To determine if PMSK1's role is dependent on its kinase activity, we used a kinase-inactive version of PMSK1, where Asp-442 was substituted with Asn (D442N) to disrupt the ATP-binding site, alongside the phosphomimetic S120D and non-phosphorylatable S120A mutations. Neither mutation affected starch content (Supplementary Fig. 19), indicating that PMSK1 fulfills the observed regulatory role on starch metabolism through its kinase activity”.

10. Page 7, line 25, Fig. 1C should be Fig. 1D.

Thank you, this has been now corrected.

AUTHOR REBUTTALS TO REVIEWERS' COMMENTS:

We thank reviewers 2 and 3 for their positive evaluation of our work. We also thank reviewer 4 for their detailed and insightful comments, which have helped us improve the clarity of our study.

A point-by-point response to the comments of reviewer 4 is presented below, in blue.

REVIEWER COMMENTS

Reviewer #2 (Remarks to the Author):

The authors have addressed the comments in a reasonable way. I have no additional comments to this revised revision

Thank you again for acknowledging the efforts we have made to improve the manuscript. We appreciate your constructive feedback and are pleased that the revisions meet your expectations.

Reviewer #3 (Remarks to the Author):

The authors fully addressed my concerns and made appropriate clarifications and revisions. The reviewer believes that the quality of the research and manuscript per se is deserved to be published in NC.

Thank you for your thoughtful feedback and support.

Reviewer #4 (Remarks to the Author):

The manuscript by Yuan et al. describes the involvement of blue-light-induced phototropin in starch metabolism in *Chlamydomonas*. The finding is quite new and interesting. However, the data and the description are insufficient to publish this paper in this journal. Authors should add some data, consider previous findings in other algal and plant species, and thoroughly revise the manuscript, in particular Introduction and Discussion sections.

Concerns

1. Authors analyzed only the regulation of GAP1 mRNA to reveal the molecular mechanism underlying phot-mediated regulation of starch accumulation, the effect of GAP1 overexpression and downregulation on starch accumulation was quite mild compared to phot and phot-kin strains. Do authors think that phot strain is defective only in starch synthesis but is normal in starch breakdown? In another Chlorophyte *Chlorella*, the blue-light-induced starch breakdown is well-known (Kowallik and Gaffron, *Planta*, Vol. 69: 92-95, 1966; Kamiya and Miyachi, *Plant Cell Physiol.*, Vol. 15: 927-937, 1974; Kowallik, *Annu Rev Plant Physiol*, Vol. 33: 51-72, 1982). It was hypothesized that this response is regulated by the 120-kDa blue-light photoreceptor localized on the plasma membrane (Matschke et al., *Photochem Photobiol*, Vol. 66: 128-132, 1997; Kamiya and Saitoh, *Physiol. Plant*, Vol. 116: 248-254, 2002). Authors compared their results with only those in *Arabidopsis stomata*, they must not ignore the results in *Chlorella* if their research focus on photosynthetic “microalgae”.

Thank you for these comments.

Concerning the effect of GAP1 expression on starch accumulation: While the 1.5-fold increase in starch content observed in *GAP1* overexpressors (*gap1-oe1* and *gap1-oe2*), which accumulated 15-fold more GAP1 mRNA than the WT (Fig. 2h), might initially appear mild; however, it is significant considering that the *phot* mutant, which expressed 80-fold more *GAP1* mRNA than the WT, exhibited a 3-fold increase in starch content, reaching ca. 15 µg starch per million cells (Fig. 2i).

This comparison highlights that the starch accumulation effect of *GAP1* overexpression is not as mild as it may seem, as much higher GAP1 expression in the *phot* mutant does not proportionally enhance starch accumulation. These findings suggest that GAP1 mRNA regulation has a meaningful impact on starch accumulation, even with moderate changes in expression levels.

We believe that the impact of *GAP1* overexpression could have been greater if the cells had been provided with higher levels of CO₂. Indeed, as shown in Rebuttal Fig. 1 (see our response to comment #7), the *phot* mutant accumulated approximately 40 µg of starch per million cells when cultures were sparged with air enriched with 2% CO₂.

It would be of course interesting to try to generate a *GAP1* overexpressor line with expression levels closer to those in the *phot* mutant (ca. 80-fold). However, achieving such high expression levels in *Chlamydomonas* can be challenging, though the intronserter approach (<https://bibiserv.cebitec.uni-bielefeld.de/intronserter>) might offer a viable solution for this purpose.

Concerning the impact of blue light in starch catabolism: Our results showed that both starch synthesis- and catabolism-related genes were upregulated in the *phot* mutant (**Supplementary Fig. 9**). Although proteomics does not necessarily capture enzyme activity, it is worth mentioning that our whole-cell proteomics data (**Supplementary Data 2**) did not reveal any statistically significant changes in the levels of alpha-amylase, the enzyme responsible for the major hydrolytic activity involved in starch degradation in *Chlamydomonas*⁴⁴. We have added a new paragraph to the discussion to compare our findings not only with *Arabidopsis* but also with *Chlorella*. This paragraph also addresses the impact of blue light on starch catabolism."

Updated and expanded text

Page 9, lines 9-19: "In the green microalgae *Chlorella*, a possible explanation for why blue light represses starch accumulation²² may lie in early studies showing that under blue light carbohydrate catabolism is enhanced^{41,42}. While the specific role of GAP1 in *Chlorella* remains to be investigated, we recently demonstrated that under red light, *Chlorella* accumulated significant starch levels even after a prolonged 7-day exposure⁴³, in accordance to earlier short-term experiments²². In *Chlamydomonas*, our results showed that both starch synthesis- and catabolism-related genes were upregulated in the *phot* mutant (**Supplementary Fig. 9**). Although proteomics does not necessarily capture enzyme activity, it is worth mentioning that our whole-cell proteomics data (**Supplementary Data 2**) did not reveal any statistically significant changes in the levels of alpha-amylase, the enzyme responsible for the major hydrolytic activity involved in starch degradation in *Chlamydomonas*⁴⁴."

2. "PHOTOTROPIN (PHOT)" means a non-functional phototropin apoprotein and thus is usually not used. When the functions of phototropins are described, authors should use "phototropin (phot)" that means phototropin holoprotein or "italicized PHOTOTROPIN (PHOT)" that means phototropin gene. Authors should use correct nomenclature of phototropins (Briggs et al., Plant Cell, Vol. 13: 993-996, 2001).

The terminology surrounding *Chlamydomonas* Phototropin is notably complex, and researchers have shown varying levels of consistency in adhering to the nomenclature guidelines initially proposed for phytochromes. In contrast, the *Arabidopsis* research community has generally maintained more consistent terminology. For instance, the publication that first described *Chlamydomonas* Phototropin (<https://doi.org/10.1034/j.1399-3054.2002.1150416.x>) adhered to the correct nomenclature. However, a subsequent publication, which included coauthors of the original discovery, abandoned this nomenclature, instead referring to the Phototropin protein as "PHOT" ([10.1111/j.1365-313X.2006.02852.x](https://doi.org/10.1111/j.1365-313X.2006.02852.x)). We acknowledge that our own publications (<https://doi.org/10.1038/nature19358>; <https://doi.org/10.1038/s41467-023-37800-6>; <https://doi.org/10.1038/s41467-023-38183-4>) have not followed the "Phot" nomenclature either. Similarly, other researchers in the field have also deviated from this terminology (e.g., <https://doi.org/10.1073/pnas.1821689116>; <https://doi.org/10.1038/s41477-018-0332-5>; <https://doi.org/10.1038/s41467-019-11989-x>). While we are open to following the reviewer's recommendation regarding nomenclature, we are concerned that this change might further contribute to confusion within the *Chlamydomonas* research community. As a result, we retained the nomenclature from the original submission, given that the other two reviewers did not comment on this issue.

3. There is no introduction of *Chlamydomonas* photoreceptors in Introduction section.

Thank you for pointing this out. It is true that we had only briefly introduced *Chlamydomonas* photoreceptors. We have now expanded our introduction as shown below:

Page 2, lines 25-32: Light is also a spatiotemporal signal; red light is detected by bilin-containing phytochromes, while blue light is perceived by flavin-containing cryptochromes and/or PHOTs, the latter characterized by two photosensory light, oxygen, or voltage (LOV) domains². *Chlamydomonas* notably lacks phytochromes in its genome³; nevertheless, this microalga has evolved a rich repertoire of photoreceptors, including a single-copy PHOT, four cryptochromes, eight rhodopsin-like proteins, and the UV-B photoreceptor UVR8⁴. Through this network of specialized photoreceptors *Chlamydomonas* regulates important cellular functions, including: gene expression, sexual life cycle, phototaxis, and photoprotection⁵⁻¹².

4. Authors should indicate what kind of statistical tests was performed in the legends of each Figure.

Thank you for pointing this out. We have now added this information for all Figures of the manuscript, as requested.

5. Supplementary Figure 1: It is still possible that decrease in starch levels could result from decrease in light intensity of red light from 50 to 45 μ E.

In designing the experiment presented in Supplementary Figure 1, we aimed to maintain a consistent fluence rate across the different light qualities (white, red, and red plus blue). The results shown in the figure indicated a repression of starch accumulation by blue light, which aligns with the findings throughout this manuscript: blue light, perceived by PHOTOTROPIN, induces PMSK1 dephosphorylation, leading to the repression of *GAPI* mRNA levels and a subsequent reduction in starch accumulation. However, we agree with the reviewer's comment that the specific effect of reducing red light intensity from 50 to 45 μ E on starch accumulation had not yet been evaluated. To address this, we conducted additional experiments and found

that starch levels at both 45 and 50 μE were statistically not distinguishable. Supplementary Figure 1 has been updated accordingly and is also presented here:

Supplementary Fig. 1. Effect of superimposing low intensity blue light on red light on starch content. (a) Starch content of WT was measured under continuous light conditions: white light, red light, or red light supplemented with low-intensity blue light. (b) Starch content of WT was also compared under the two different red-light intensities used in (a). Light intensities in $\mu\text{mol photons m}^{-2} \text{s}^{-1}$ are indicated in the x-axis.

6. Furthermore, it is not clear whether the suppression of beneficial effect of red light was specific to blue light and which wavelength of light is effective in starch accumulation under white light condition. Authors should analyze starch levels in cells exposed to only blue light and other wavelength light (green and far-red).

In this study, we adopted a hypothesis-driven approach to demonstrate that blue light acts as a repressor of starch accumulation in *Chlamydomonas*. Using a genetic approach, we showed that this repression is mediated via PHOTOTROPIN (PHOT) by comparing different mutants lacking blue light receptors. Our discovery-driven approaches further strengthened this hypothesis.

First, whole-cell proteomics revealed that GAP1 is a key player in the regulation of starch levels in this alga and that its expression is controlled by the kinase activity of PHOT. Second, phosphoproteomics identified PMSK1 as a key regulator of starch accumulation mediated by blue light. Phosphomimetic analyses confirmed that the residue S120 of PMSK1 is critical in this regulation: the phosphorylated form activates GAP1 and promotes starch accumulation, while PHOT-mediated dephosphorylation of S120 leads to the inactive form of PMSK1, ultimately repressing starch accumulation.

Taken together, our findings provide compelling evidence that blue light, via PHOT, represses starch accumulation in *Chlamydomonas*. The seemingly beneficial effect of red light can be attributed to the absence of blue light (the repressor) in monochromatic red light conditions.

Consequently, our work focused on elucidating the PHOT-PMSK1-GAP1 pathway that links blue light to starch regulation.

Additionally, we proposed that other blue-light-responsive proteins might independently inhibit starch accumulation or regulate GAP1 outside of the PHOT-PMSK1-GAP1 pathway.

This hypothesis is based on the following observations:

(i) *phot* mutants accumulate more starch under red light (Fig. 1d), despite consistently high GAP1 mRNA levels across light conditions (Fig. 2d).

(ii) Red light enhances GAP1 mRNA levels in the *phot-kin* strain (Fig. 2g).

Throughout our study, we systematically compared the effects of blue light alone (repression active), white light (which includes both the repressing blue light and red light), and red light alone (lacking the repressing blue light). Exploring the effects of green or far-red light was deemed outside the scope of this work, as our primary focus was on dissecting the blue-light-mediated repression of starch accumulation.

7. Because authors analyzed only blue-light effect, the involvement of channelrhodopsins in starch accumulation remained to be determined.

In response to the reviewer's comment, we performed additional experiments including the double *chR1chR2* mutant, which lacks both Channelrhodopsin 1 and 2, alongside the WT and *phot* strains. These experiments were conducted under low intensity continuous white light (20 $\mu\text{mol photons m}^{-2} \text{s}^{-1}$) and aeration with air enriched with 2% CO₂, in contrast to other experiments where no CO₂ enrichment was applied.

8. Page 3, lines 25-29

Authors should mention not only very thick starch sheaths around pyrenoid but also the ectopic starch accumulation apart from pyrenoid in phot mutant, because authors put two arrows on a photograph for phot mutant in Fig. 1C. Because there is no introduction of starch accumulation pattern in *Chlamydomonas* in Introduction section, most readers would not understand it. At least, I do not know what is the difference between WT and phot in Supplementary Fig. 4.

Thank you for this valuable comment. We have carefully addressed your concern and revised the text in the manuscript as follows:

Page 3, lines 39-43 and page 4, lines 1-2: “The accumulation of high levels of starch in the phot mutant was also confirmed by transmission electron microscopy (TEM), revealing striking changes in starch deposition patterns compared to the wild type. As indicated by as the arrows in the TEM images in **Fig. 1c**, the chloroplasts of the phot mutant are filled with starch granules and its pyrenoid, is surrounded by abnormally thick starch sheaths. In contrast, in the chloroplasts of the WT and phot-C1 strains starch is predominantly localized as thin sheaths around the pyrenoid (**Fig. 1c, Supplementary Fig. 4**).”

We also added some missing information in the introduction section:

Page 1, lines 19-24; new text is shaded in grey: “Photosynthetic microalgae convert light into chemical energy in the form of ATP and NADPH, which fuel CO₂ fixation in the Calvin–Benson cycle. This process of CO₂ fixation is initiated by the activity of the CO₂ -assimilating enzyme ribulose 1,5-bisphosphate carboxylase/oxygenase (Rubisco)¹. In eukaryotic algae, such as the model photosynthetic green microalga *Chlamydomonas reinhardtii* (hereafter *Chlamydomonas*), concentrated CO₂ is delivered to Rubisco within a specialized microcompartment in the chloroplast called the pyrenoid ².”

Page 1, lines 33-44; new text is shaded in grey: Fixed CO₂ in combination with nitrogen is used to synthesize amino acids—the building blocks of proteins that drive biochemical reactions. It is also employed in the synthesis of cellular reserves that ensures carbon and energy

supply during waning period. The most abundant carbon reserve in the model photosynthetic green microalga *Chlamydomonas* is starch; its synthesis occurs during the day and its degradation is triggered at night to sustain energy-demanding cellular functions¹⁴. Starch also serves another critical role as starch sheaths encasing the pyrenoid act as a barrier, reducing CO₂ leakage from this structure¹⁵. While actively dividing, *Chlamydomonas* cells accumulate starch mostly around the pyrenoid. In contrast, when subjected to a stress such as nutrient limitation, starch granules are massively accumulating in the chloroplast stroma¹⁶. Little is known about the molecular mechanisms underlying the control exerted by light on carbon storage in microalgae, and current knowledge is limited to factors impacting starch accumulation under adverse environmental conditions, such as: nitrogen¹⁷ and phosphorus¹⁸ limitation.

9. There is no data for phot-C1 strain in Supplementary Fig. 4 although that was mentioned in the text (line 28).

Supplementary Fig. 4 shows that the TEM pattern shown in Fig. 1c was consistently observed across the cell population. We thank the reviewer for pointing out that those data were missing for *phot-C1*. We have now added the relevant pictures in the revised Supplementary Fig. 4. We also decided to move the TEM of *WT/pmsk1^{S120D}* and *phot/pmsk^{S120A}* from Supplementary Fig. 4 to Supplementary Fig. 16.

10. Page 5, lines 25-36

I have no idea why authors focused on cluster E. “To identify missing components in the phototropin-mediated signalling pathway”, we will usually screen the direct substrate of phototropins with phosphoproteomics and thus focus on clusters M and O. At least, the rationale that authors focused on cluster E should be described here.

We agree that to identify direct substrates of PHOT, as PHOT is a kinase, the primary focus would typically be on Clusters M and O, which contain PHOT and blue-light-dependent phosphopeptides. However, our objective in this study was to identify missing components in the PHOT-mediated signaling pathway, even if they are not direct substrates of PHOT. Currently, no substrates or components of any CrPHOT signaling pathway have been identified or reported in *Chlamydomonas*.

We currently are exploring a diverse range of promising candidates from clusters E, M, and O. Among the candidates, PMSK1 stood out because of its strong homology to CBC1 and HT1-well-characterized substrates of PHOT in *Arabidopsis*. Furthermore, our data showed that residue S120 of PMSK1 undergoes PHOT-dependent dephosphorylation, which provided compelling evidence for selecting PMSK1 for further investigation.

In the manuscript, we prioritized uncovering the functional role of PMSK1 before addressing its evolutionary connection to CBC1 and HT1 in *Arabidopsis*. We believe this approach aligns well with the overarching narrative of our study, which focuses on linking blue-light signaling to starch accumulation through the dissection of the PHOT-mediated signaling pathway. Any initial questions the reader might have about why PMSK1 was selected are resolved later in the manuscript, as we transition to discussing its evolutionary connection to CBC1 and HT1 in *Arabidopsis*.

11. Page 6, lines 3-5

In Fig. 3c and d, the unphosphorylated band can be observed 20 min after blue light irradiation in phot strain. The band is very weak, but significant compared to red-light irradiated samples. Authors should revise the sentence as follow; whereas in the phot background, dephosphorylation was hardly observed under blue light (Fig. 3c). No dephosphorylation occurred under red light in both strains (Fig. 3c).

Thank you for pointing this out. We have made the necessary changes, and the text now reads: “In the WT background, dephosphorylation occurred only under blue light (Fig. 3c). In contrast, in the *phot* background, dephosphorylation was barely observed, even under blue light (Fig. 3c). Furthermore, no dephosphorylation was observed under red light in either strain (Fig. 3c).”

12. The data in Supplementary Fig. 16 is very important and thus should be transferred to Fig. 3.

This a good suggestion, thank you. We replaced the graph showing the kinetics of PMSK1 dephosphorylation (previously Fig. 3d) with Supplementary Fig. 16, which is now Fig. 3d.

13. Page 6, lines 12-38

Authors should refer to Supplementary Fig. 4 here.

In response to comment 9 we decided to move the TEM of *WT/pmsk1^{S120D}* and *phot/pmsk^{S120A}* from Supplementary Fig. 4 to Supplementary Fig. 16. This last one is properly cited in the text that the reviewer points out.

14. Page 7, lines 10-34

This part should be transferred to Discussion section.

We appreciate the reviewer’s suggestion to move this section to the Discussion. However, we believe this paragraph serves an important purpose in the Results section. It provides a concise summary of the manuscript’s data and effectively introduces Figure 6 by presenting all the relevant details that underpin the proposed model. This integration allows us to comprehensively present our findings while maintaining a logical flow in the narrative. Therefore, we prefer to retain this paragraph in the Results section.

15. Page 8, lines 1-2

Our work establishes PMSK1 as the first kinase described to mediate phot-dependent signalling in “Chlamydomonas (or green alga)”

We have modified the text in response to this comment as follows: “Our work establishes PMSK1 as the first kinase described to mediate PHOT-dependent signalling in algae”.

16. Page 8, lines 13 -14

References 30 and 31 are not papers that indicate that “NPH3 may function as a substrate adaptor in a CULLIN3-base E3 ubiquitin ligase complex, targeting phot1 for

ubiquitination. Moreover, the role of NPH3 in phot1 ubiquitination is controversial in the field of phototropin research. Thus, the sentence should be deleted.

Thank you for this comment. We have removed this sentence.

17. Page 8, lines 14 -15

This sentence leads to misunderstanding. Completely dephosphorylated NPH3 is inactive and partially phosphorylated NPH3 is active (Christie et al., Plant Physiol, Vol. 176: 1015-1024, 2018).

We thank the reviewer for this comment. We have removed the sentence that could lead to misunderstanding and have instead focused on using this as an example of how Phot1, a kinase, may be involved in the dephosphorylation of a downstream effector. This provides a parallel to how *Chlamydomonas* PHOT could be required for the dephosphorylation of the downstream effector PMSK1. The revised text now reads:

Page 8, lines 25-28: “In *Arabidopsis*, which encodes two phototropins, Phot1 and Phot2, blue light induces the phosphorylation of NPH3 (NON-PHOTOTROPIC HYPOCOTYL3) at serine 744 (S744) in a Phot1-dependent manner. This phosphorylation creates a 14-3-3 binding site enabling NPH3 to associate with 14-3-3 proteins. Subsequently NPH3 gets dephosphorylated^{35,36}.

18. Page 8, lines 21-25

Reference 33 is a paper on CBC1/2, but not for HT1. Authors should cite an appropriate paper on HT1.

While this reference (DOI: 10.1038/s41467-017-01237-5) primarily focuses on CBC1 and CBC2, it also demonstrates their interaction with and phosphorylation by HT1, as highlighted in their study. We acknowledge the reviewer’s point regarding the omission of a citation for the discovery of HT1. In response, we have now added the appropriate reference (DOI: 10.1038/ncb1387) in the revised text.